# Disentangling the role of floral sensory stimuli in pollination networks

Aphrodite Kantsa [1,6], Robert A. Raguso[2], Adrian G. Dyer[3,4], Jens M. Olesen [5], Thomas Tscheulin[1] & Theodora Petanidou [1]

Despite progress in understanding pollination network structure, the functional roles of floral sensory stimuli (visual, olfactory) have never been addressed comprehensively in a community context, even though such traits are known to mediate plant–pollinator interactions. Here, we use a comprehensive dataset of floral traits and a novel dynamic data-pooling methodology to explore the impacts of floral sensory diversity on the structure of a pollination network in a Mediterranean scrubland. Our approach tracks transitions in the network behaviour of each plant species throughout its flowering period and, despite dynamism in visitor composition, reveals significant links to floral scent, and/or colour as perceived by pollinators. Having accounted for floral phenology, abundance and phylogeny, the persistent association between floral sensory traits and visitor guilds supports a deeper role for sensory bias and diffuse coevolution in structuring plant–pollinator networks. This knowledge of floral sensory diversity, by identifying the most influential phenotypes, could help prioritize efforts for plant–pollinator community restoration.

[1] Laboratory of Biogeography and Ecology, Department of Geography, University of the Aegean, 81100 Mytilene, Greece. [2] Department of Neurobiology and Behavior, Cornell University, Ithaca, NY 14853, USA. [3] Department of Media and Communication, Royal Melbourne Institute of Technology, Melbourne, 3000 VIC, Australia. [4] Department of Physiology, Monash University, Melbourne, 3800 VIC, Australia. [5] Department of Bioscience, Aarhus University, Ny Munkegade 116, 8000 Aarhus, Denmark. [6]Present address: Department of Environmental Systems Science, ETH Zürich, Schmelzbergstrasse 9, 8092 Zürich, Switzerland. Correspondence and requests for materials should be addressed to A.K. (email: a.kantsa@gmail.com)

Early in the twentieth century, biologists adopted network theory in order to investigate complex systems such as food webs[1]. Networks now constitute a powerful analytical tool used in the study of mutualistic interactions, such as pollination[2], seed dispersal[3], plant–mycorrhiza associations[4] and cleaner–client reef fish relationships[5]. In particular, network analysis has transformed the conceptualization of pollination from a biological function traditionally studied in isolated pairs or small groups of species, to a key ecosystem function that sustains primary productivity and the stability of communities[1]. At present, given that all the major worldwide threats to biodiversity affect plant–pollinator (p–p) interactions as well[6], intensive global research efforts focus on understanding the structure and dynamics of p–p networks, so that conservation and restoration strategies can be effectively employed[7,8].

During the last twenty years, there has been a lively debate regarding the interplay of ecological and evolutionary specialization in p–p interactions. The assessment of large interaction datasets revealed that symmetrical specialization in communities is rarer than expected from the pollination 'syndrome' paradigm[9], the applicability and influence of which remain contentious subjects[10,11]. Tight pairwise or 1:1 species relationships strictly matching the 'syndrome' concept (according to which specific sets of plant traits provide some predictive power in the identification of evolutionarily important pollinator groups) are indeed rare in natural communities[12]. Thus, studies of p–p networks have tended to de-emphasize specialization and, as a consequence, the importance of floral phenotypes as adaptations in ordering network structure. Long-term observations of p–p networks indicate that they are characterized by a great temporal plasticity of interactions[2,13]. Furthermore, the opportunistic foraging behaviour of flower-visiting insects has been attributed to extreme seasonality in certain biomes[2], whereas theoretical models have inferred that, in mutualistic networks, linkage rules may largely be explained by phenology and/or species' abundance[14–16].

However, we know that, in pollination networks, nearly half of the visitor species interact with only one or very few plant species[13,17]. Moreover, mutualistic interactions entail at least some degree of phenotypic complementarity[18], exemplified by the simple matching between proboscis length and corolla depth[19,20], to the more sophisticated interplay of floral stimuli (e.g. scent, colour) and the sensory systems of pollinators[21,22]. The shared evolutionary history of plants and pollinators is linked to their interaction patterns[23–25], is considered a major driver of floral diversification and, although it may not act alone[26], has been shown to operate within the almost generic asymmetrical structure of mutualistic networks[18,27]. Recently, our study revealed a phenotypic integration between floral colour (as perceived by pollinators) and scent at a community level among the flowering plants[28]; this finding suggests a coordinated adaptation of plants to the sensory systems of pollinating insects. We build upon this finding by asking whether floral phenotypes that match visitors' physical and sensory biases in a community context represent evolutionary vestigial traits or relics with no extant function, or alternatively, whether they are correlated with the realized pollinator-niches of the plants.

Distinct sensory biases and cognitive abilities of pollinators are expected to filter the information available in floral landscapes[28], and to shape foraging behaviours[29]. In this context, insect responses to natural floral volatile blends were found to correlate with visitation patterns[30], and experimental manipulation of those blends was shown to reversibly affect visitation patterns in two keystone plant species studied simultaneously in a community[31]. Similarly, floral reflectance has been connected to visitation patterns[32,33], and recently, the phenotypic matching between the flower colour of temporally overlapping plants and the visual systems of pollinators was shown to influence insect attraction and the plants' ecological specialization in one community[34]. However, pollinators likely experience floral traits as multimodal sensory information, rather than solely visual or olfactory cues[35]. Pollination network studies have not yet considered the full complement of floral stimuli; hence, it remains unclear whether community visitation patterns have a sensory basis or, as inferred in most studies to date, they represent a function of floral density and/or phenological matching. Answering this question requires a comprehensive dataset on the sensory ecology of an entire floral community, including human-unbiased parameterizations of floral scent and colour[28], and an evaluation of specialization through a balanced design that directly compares the relative impacts of phylogeny, floral traits, phenology and density.

Species must co-occur in time in order to interact; therefore, flowering phenology is the first factor filtering p–p interactions[2,13,16,32,36] in a community. Yet, by assessing the overall (static) p–p network of the entire flowering/sampling season, one considers putative interactions that are not possible in time, thereby inflating the calculated measures of specialization[37]. The resulting inflation can be considerable, given that most species in many communities, including the Mediterranean ones, tend to have short flowering phenophases[2,13,38]. Recent studies have addressed this issue by partitioning the total network into smaller regular time-sequential networks[16,39]. However, even network studies with a day-scale resolution inevitably employ the static version of plants' ecological specialization and centrality (i.e. the degree of the influence that a given species has on the network's structure)[39], simply because this type of data aggregation provides the total number of links among the nodes, against which the realized interactions will be compared.

To overcome this methodological bias, we introduce a dynamic approach to data pooling in bipartite ecological networks, the 'phenonet' structure, which partitions the total cumulative network into a set of networks equal in number to the interacting species in the community. Here, a phenonet becomes a snapshot of the entire p–p network of the community, encompassing the interaction spectrum occurring within each species' phenophase. Our method is based on the simple facts that (i) not all species interact directly with each other, but only with the temporally co-occurring species and (ii) species interact directly with other trophic levels only during their phenophase. Thus, the actual ecological or evolutionary specialization may vary according to time and depending on the ecological context, i.e. the antagonistic or facilitative interactions with co-occurring species.

The second key element known to determine interaction patterns and the structure of networks is floral density[19,34]. In general, floral abundance is used in p−p network studies as an independent factor; however, from a functional ecological standpoint, the number of flowers represents the relative abundance of specific (e.g. visual and olfactory) phenotypic traits that constitute the floral sensory landscape in a community of blooming plants. We thus adopt the concept of 'apparency' used in chemical ecology of herbivory[40] to describe the multiple ways that a plant can present olfactorily or visually distinguishable (and thus more apparent) flowers across the community.

Here we investigate the role of floral sensory stimuli on the structure of a p–p network in the phrygana, a natural Mediterranean scrubland, by employing the above-mentioned dynamic data pooling, and by accounting for species' functional abundance and phylogeny. We specifically ask (i) whether plant centrality and generalization are significantly associated with floral phenotype, (ii) whether floral phenotype predicts visitation rates by the different pollinator groups and (iii) which phenotypic elements are important for the structure of the p–p network. Our findings

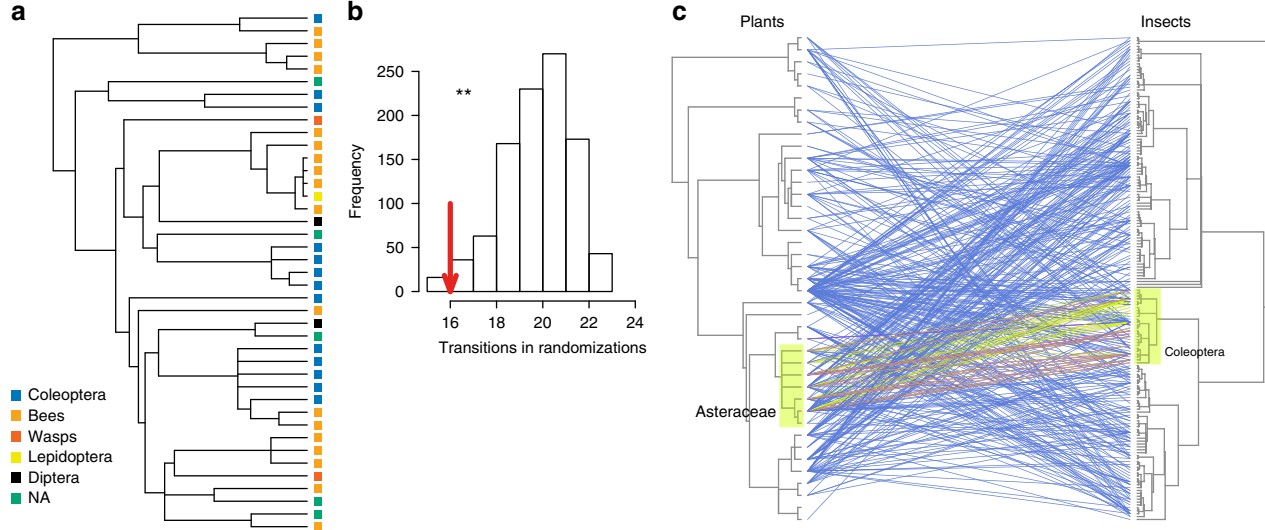

**Fig. 1** The phylogenetic structure of plant–pollinator interactions. **a** The prevailing pollinator group mapped, as a categorical variable, within the phylogeny of the plants. NA: ≥2 groups with the highest visitation rate. **b** The histogram of the null model of evolutionary transitions according to the algorithm 'phylo. signal.disc' for discrete traits. The arrow shows the number of observed transitions of the trait, which is significantly lower than the randomized median (**$P = 0.008$). **c** The two clades showing the lowest residuals (highest phylogenetic congruence) in the Procrustean cophylogenetic analysis. For details, see Methods

reveal that floral scent and colour as perceived by pollinators are related to plant behaviour in the community; species that are critical for holding together the entire network are characterized by specific floral traits. Moreover, the visitation rates by the different pollinator groups are correlated with different floral stimuli. Overall, our approach, which can be applied in different bipartite networks, disentangles the significance of floral phenotype in the interacting community, and helps to define the most influential plant species.

## Results

**The static p–p network of the community.** The total p–p network (Supplementary Data 1)[41] sampled in two consecutive flowering periods (see Methods) consisted of 41 plant species (18 families) and 168 insect species (44 families), including 403 plant–insect species links and 7955 interaction events, i.e. insect visits to individual flower units. Most of these events were made by bees (57.9%), followed by beetles (Coleoptera; 33.8%), flies (Diptera; 5.2%), butterflies (Lepidoptera; 2.2%) and wasps (0.7%) (Supplementary Data 1)[41]. The bees were also the most species-rich group (66 taxa), followed by flies (37), beetles (26), wasps (19) and butterflies (17) (Supplementary Table 1). Connectance (i.e. the number of realized interactions as a proportion of all possible interactions) was 5.9%, which is an intermediate value for Mediterranean scrubland networks[2,19]. Nestedness (i.e. the asymmetrical pattern in which interactions of specialist species are subsets of the interactions of the generalist ones, with the latter representing the core of interacting species in a community) was 95.5%, a value typical for mutualistic networks[17].

The coupled evolutionary history of plants and insects was associated with the structure of the pollination network in the community. The cophylogenetic signal of the binary network (including only the presence/absence of p−p interactions) was significant (Procrustes sum of squares, $m^2 = 0.97$, $P = 0.001$), indicating that the backbone of interactions was correlated with the shared evolutionary history of plants and pollinators. For example, interactions between the sunflower family (Asteraceae) and beetles (Coleoptera) showed the lowest Procrustean residuals, implying particularly strong cophylogenetic signal between these

clades compared with the rest of the network (Fig. 1c). Given that the Procrustean analysis employs the binary version of the network (i.e. not including the number of visits for each link)[42], we could suggest that it actually measures the phylogenetic signal of the initial attraction between plants and insect, exactly because it equally values all links: some p−p links occurred repeatedly (i.e. many visitation events), yet some others only occurred once because although the insect was attracted, it decided not to visit another flower of the same plant species. Thus, this approach revealed that the "first contacts" between plants and pollinators are related to p−p coevolution.

Moreover, we found that the modular structure of the network[43] (consisting of nine modules, Supplementary Table 2) is significantly associated with the insect phylogeny (Rezende's algorithm, $P < 0.001$), and not with that of the plants ($P = 0.355$). The fact that the insect (instead of the plant) phylogeny shapes module composition could imply convergence of floral traits of unrelated plant lineages as a response to selection pressure by specific clades of pollinators[28], or the exploitation of specific floral resources by the different taxonomic groups of pollinators from randomly selected plant species (refs. [24,44] and refs. therein).

Finally, the phylogenetic relatedness of plants was a source of variation for the prevailing group of visitors. When visitation by each major pollinator group was treated as a categorical plant variable, we found that for each plant species in the community, the predominant group (as to no. of visits), showed a significant phylogenetic signal, i.e. phylogenetically related plant species were mainly visited by the same pollinator group (Rezende's algorithm, $P = 0.008$; Fig. 1a, b). Furthermore, significant phylogenetic signals were found for the visitation rates by flies, beetles, and wasps (Supplementary Fig. 1). These associations align with the view that the phylogenetic congruence of interacting plants and pollinators occurs at high taxonomic levels (orders, rather than species)[24], and suggest that grouping the flower visitors of the Mediterranean scrubland studied according to their taxonomic order was ecologically meaningful.

**Phenonets reveal the temporal plasticity of interactions.** A phenonet is the interaction network covering each plant species'

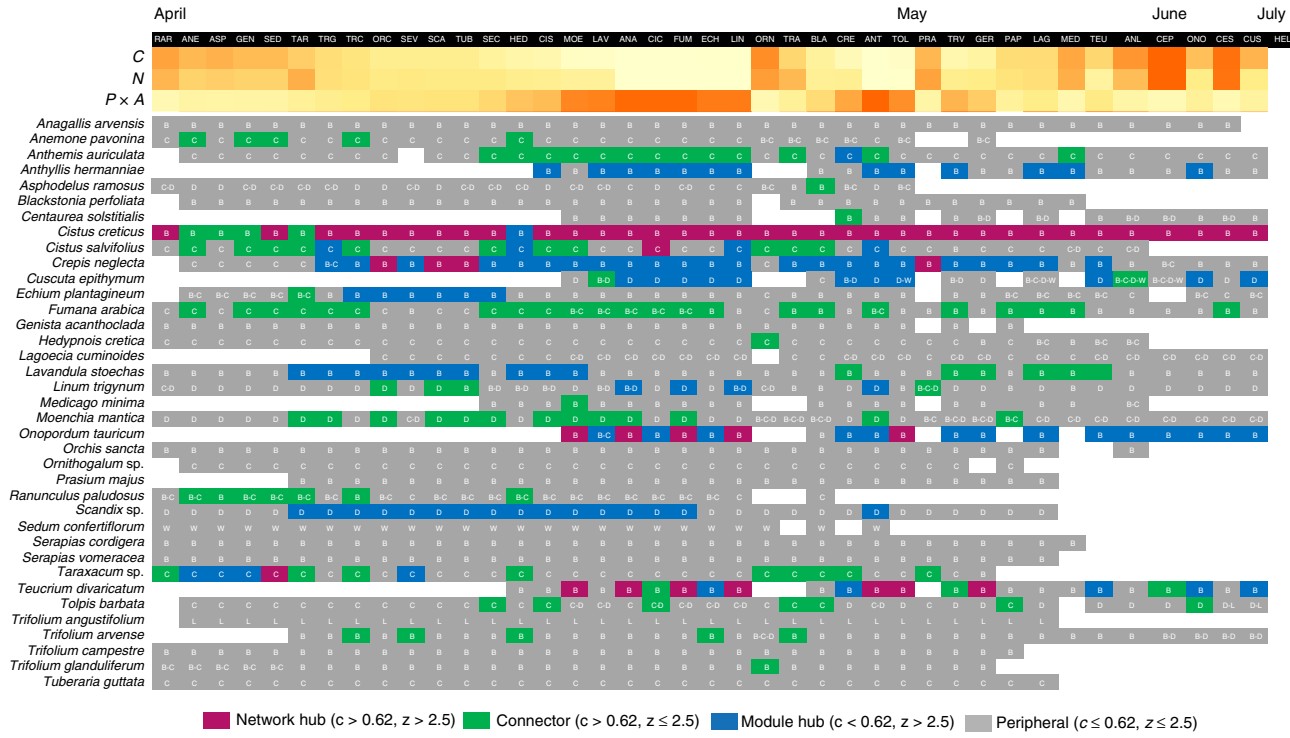

**Fig. 2** The functional roles of the plant species in the phenonets of the community. Plant phenonets are shown in the top row (shaded in black) and are sorted temporally, so that plants with earlier start and shorter duration are ordered first, from left to right. Months correspond to the start of flowering only. For each phenonet, the values of three network properties are provided in a gradient colour scale: Connectance (C), Nestedness (N), and Network size (P × A). For each plant species (rows), the pollinator group with the highest number of species visiting the plant in each phenonet is also provided: Bees (B), Coleoptera (C), Diptera (D), Lepidoptera (L) and Wasps (W). White cells denote absence from the respective phenonet. For the phenonet name abbreviations, see Supplementary Table 6

flowering period. The plant phenonets in the study community (Supplementary Table 3) had various sizes depending on the phenophase length of each plant species. The smallest phenonet consisted of 42 insect and 15 plant species (phenonet of *Centaurium pulchellum*), whereas the largest phenonet contained 165 insect and 36 plant species (*Anthemis auriculata*). One caveat is that this data-pooling method generates binary networks, and hence the metrics and algorithms used in the analysis should be appropriately selected (see Methods).

Nestedness was negatively correlated with plants' phenonet size (Pearson's $r = -0.82$, $P < 0.001$, $N = 40$) (Fig. 2), and ranged from 78.7% (*Ornithogalum* sp.) to 95.8% (*Echium plantagineum*, *Linum trigynum*). For plant species that flowered at the temporal extremes of the flowering season, phenonets became smaller in size, more connected and more nested compared with the mid-season ones, when the interaction patterns were more random (Fig. 2). Nestedness in mutualistic systems is directly linked to community dynamics. Specifically, it has been shown to reduce effective competition and enhance the number of coexisting species[45], and to be positively associated with community persistence in seasonal or unpredictable environments[46]. The connectance of plant phenonets ranged from 6.6% (*Anthemis auriculata*) to 19.8% (*Centaurium pulchellum*), and also was negatively correlated with the size of the phenonet (Pearson's $r = -0.88$, $P < 0.001$, $N = 40$).

The phenonet approach yields significantly higher ecological generalization and centrality values of the plant species within a community, compared with the static p–p network. We compared three fundamental node properties, describing the centrality and generalization level of plant species in the total p–p network and in the phenonets: (i) Normalized Degree (ND), (ii) Betweenness Centrality (BC) and (iii) Closeness Centrality (CC).

ND is the size of the pollinator niche of the plants in the community; BC is the extent to which a plant species connects parts of the network that would otherwise be isolated; CC measures how close one plant species is to the other co-flowering ones via shared pollinators[47]. The mean values of the three node properties examined were all significantly higher in the species' phenonets than in the static network (Fig. 3; Supplementary Table 4). Specifically, mean ND was 68.8% higher in the phenonets than in the static network, mean BC was 37.0% higher, and mean CC was 33.3% higher ($N = 37$). The functional or topological roles of the plants (i.e. peripherals, module hubs, connectors and network hubs)[43] also differed between the static network vs. the phenonets (Supplementary Table 4; Fisher's exact test for count data, $P < 0.001$, $N = 37$). Specifically, six species acquired more generalist topological roles (three peripheral species became connectors, two peripherals became module hubs, one module hub became a connector), and conversely, three species acquired more specialist roles (two connectors became peripherals, and one module hub became a peripheral; Supplementary Table 4).

The series of phenonets allowed for a detailed observation of the p–p network's temporal dynamics. During the entire sampling season, the aforementioned functional roles of plants[43] were generally far from static: species considered peripheral (i.e. extreme specialists) in the static network temporarily become generalists (e.g. *Anemone pavonina*, *Ranunculus paludosus*; Fig. 2), in the phenonets of other species. Apart from *Cistus creticus*, which was the only network hub in the static network (Supplementary Table 4), other species also served as network hubs for shorter periods, e.g. *Cistus salviifolius*, *Crepis neglecta*, *Onopordum tauricum* and *Taraxacum* sp. (Fig. 2). On the other hand, there were specific plants that maintained their role as

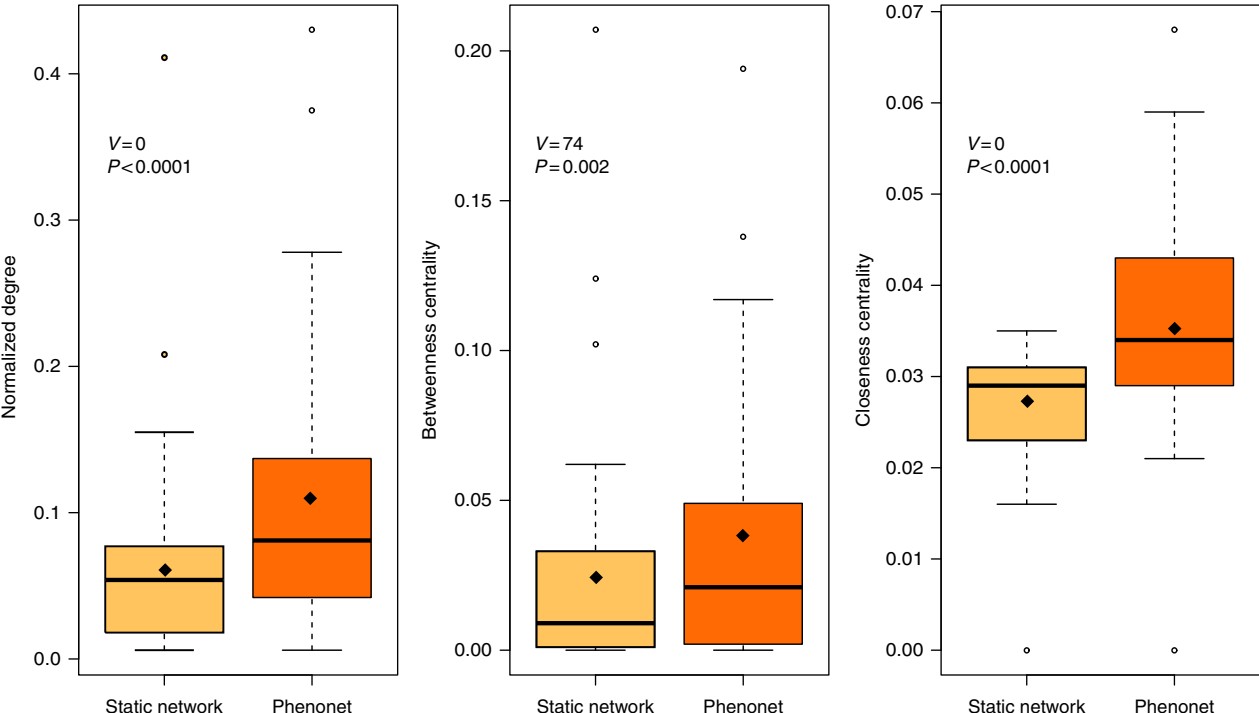

**Fig. 3** Plant species' behaviour as estimated in their phenonets vs. the static network. The three species-level network properties tested are the Normalized Degree (ND), Betweenness Centrality (BC) and Closeness Centrality (CC) of the study plants (for details, see Methods). Statistical significance was assessed with paired Wilcoxon signed-rank tests ($N = 37$)

module hubs, thus their pollinator "clientele" persisted in the community over a long time, across several plant phenonets (e.g. *Anthyllis hermanniae*, *Crepis neglecta*, *Lavandula stoechas* and *Scandix* sp.). Interestingly, the above-mentioned hubs mainly attract bee species (Fig. 2). 'Connectors' (i.e. species that connect different modules) show also a general constancy regarding the most species-rich order of visitors they receive. Thus, even generalist plants or those with shifting roles during the flowering season are mainly associated with species of specific insect orders. In sum, our approach reveals (i) the multidimensional temporal mosaic of the functional diversity of insect-pollinated plants, (ii) species that are highly influential in the community during their phenophases and (iii) those plants that maintain in time certain modules of the p–p network, being thus potentially important for the conservation of their module partners, both plants and animals.

**The role of floral phenotype**. After evaluating the analytical adequacy of the phenonets, and revealing the temporal dynamics of the interactions, we tested the three network node properties that describe the plant species' behaviour within them (ND, BC and CC) against the components of floral phenotype. For this, we applied generalized least-squares models corrected for statistical dependence due to phylogenetic proximity (PGLS in Table 1). Apart from the standard floral traits (viz. scent variables, colorimetric properties, symmetry, nectar presence and corolla depth—see Methods), we calculated the chemical and the physical floral apparency for each plant species (Fig. 4). These values describe the degree to which a plant species has more or less noticeable flowers compared to its co-flowering species, regarding its inflorescence emissions or its physical appearance, given its floral abundance in the community (Supplementary Fig. 2). We found that the behaviour of the plant species in their phenonets was significantly associated with floral sensory stimuli (Table 1).

The level of generalization (ND) was significantly related to visual (viz. trichromatic floral colour, horizontal and vertical apparency) and chemical floral traits (viz. apparency of sesquiterpene and aliphatic volatiles). BC was significantly related to trichromatic colour, sesquiterpene apparency, proportion of aliphatics in the emissions and surface apparency. Thus, the trichromatic floral colour, scent and surface apparency predicted the importance of plant species for the cohesion or, inversely, the fragmented nature of the p–p network. Interestingly, CC, the rate of shared pollinator species with other plants in the community, was correlated only with floral scent properties (Table 1). CC values increased with the number of pollinator species that a plant species shared with other plants, i.e. with the pollinator niche overlap[48]. Thus, pollinator niche similarity was significantly associated with floral scent in our focal community. This result expands upon recent findings about the importance of floral scent as a key partitioning factor for two major plant nodes within a p–p network[31]. Since high values of BC and CC identify species with the greatest influence on network topology[48], we may conclude that the diversity of floral sensory stimuli, i.e. the relative variability of colour descriptors and scent properties of the flowers, was a driver of plant functional complexity in the assembly.

Then, we tested the relationship between the different components of floral phenotype and the overall structure of the static p–p network. For this, we applied a multivariate-response generalized linear model designed for compositional data (MGLMs in Table 2), using the weighted (quantitative) static network interaction matrix as a multidimensional response variable (see Methods). We found that visual and olfactory floral stimuli (viz. trichromatic colour, emissions of sesquiterpene volatiles and horizontal apparency) contributed significantly to the network link structure, implying that interactions were not randomly assembled (Table 2). Repeating the same analysis for every subset of the p–p network including a major pollinator taxonomic group (viz. bees, wasps, Coleoptera, Diptera and

**Table 1 Floral traits associated with the three node properties of plants in the network studied**

| Node property | AIC$_{null}$ | AIC$_{best}$ | Predictor | d.f. | F | P-value | Slope |
|---|---|---|---|---|---|---|---|
| Normalized Degree | −4.1 | −48.1 | Trichromatic colour ($r_{tri} \times \theta_{tri}$) | 6,30 | 8.11 | **0.008** | + |
| | | | %Aliphatics | | 41.85 | **<0.001** | − |
| | | | Sesquiterpene apparency | | 16.10 | **<0.001** | + |
| | | | Height apparency | | 23.02 | **<0.001** | + |
| | | | Surface apparency | | 12.93 | **0.001** | + |
| | | | Normal boiling point of scent blend | | 4.06 | 0.053 | + |
| Betweenness Centrality | −5.0 | −36.3 | Trichromatic colour ($r_{tri} \times \theta_{tri}$) | 4,32 | 6.70 | **0.014** | + |
| | | | %Aliphatics | | 17.02 | **<0.001** | − |
| | | | Sesquiterpene apparency | | 24.66 | **<0.001** | + |
| | | | Surface apparency | | 12.33 | **0.001** | + |
| Closeness Centrality | −119.7 | −123.6 | %Aliphatics | 2,34 | 5.52 | **0.025** | − |
| | | | Al_app × Be_app × Mo_app × Se_app | | 2.62 | 0.115 | + |

The best phylogenetically informed generalized least-squares models according to AIC are presented. The AIC value of the null model (-1) is provided for comparison. Significant P-values (>0.050) are indicated in bold. Positive or negative relationships are given (slope +: >0, –: <0). For details on the predictors, see Methods and the Supplementary Information
Al aliphatics, Be benzenoids, Mo monoterpenes, Se sesquiterpenes, app apparency

Lepidoptera) and their host plants revealed that sesquiterpene emissions were associated with the link structure in the networks of bees, whereas physical apparency was important for the networks of almost all groups of insects (Table 2). The importance of physical apparency (see also ref. [32]) might reflect a general attractiveness response or opportunism by insects, related either to optimal foraging strategies (e.g. in honeybees) or to limited flight capabilities (e.g. in beetles).

Regarding the chemical phenotypic traits, we found that higher sesquiterpene emissions characterized plants with broader pollinator-niches and stronger impact on the cohesion of the network. These semi-volatile compounds are typical of the so-called "aromatic plants", which release high amounts of VOCs from glandular trichomes on their flower and leaf surfaces[49], in the Mediterranean-type scrublands. At the same time, such plants represented central nodes in the study network: *Lavandula stoechas*, *Teucrium divaricatum* and *Cistus creticus* (Fig. 4) were abundant perennials with high visual and chemical floral apparency, emitting high amounts of terpenes (Figs. 3–4; Supplementary Data 2 (ref. [41]); Supplementary Table 5). Given the risk of low efficacy of volatile emissions due to the strong winds and high temperatures in the phrygana[28,50], the functional roles of the aromatic Mediterranean plants deserve further scrutiny. Compared with other classes of floral volatiles, sesquiterpenes are larger and less volatile molecules[51], allowing us to suggest that the heavy emissions of these compounds create a compact olfactory environment that is sufficiently resistant to dilution by wind forces and to rapid volatilization[50,52]. Considering sesquiterpenoid emissions as an adaptation to the abiotic environment[28,35] remains to be tested in the Mediterranean-type biomes of the world, characterized by the abundance of such plants (e.g. Californian chaparrals and South African fynbos). The emission of aliphatic compounds (i.e. non-terpenoid compounds that lack ringed C-skeletons but otherwise vary substantively in chain length and degree of saturation) demonstrated unique associations. Although they were negatively correlated with BC and ND (Table 1), implying that they characterize specialist plants in the network, they were positively associated with bee and butterfly visitation rates (Table 3). Given that many of these compounds can show behaviour-eliciting functions and are highly attractive to certain insects (e.g. male andrenid bees[53]), the observed pattern in the phrygana probably indicates that aliphatic volatiles mediate specialized relationships with bees or butterflies. Manipulative experiments will be necessary to distinguish between direct functions (e.g. pollinator attractants)

or indirect effects (e.g. pleiotropic interactions with other traits) of aliphatic compounds in such interactions.

Visitation rates by the other pollinator groups were associated with floral sensory stimuli (Table 3) with the exception of wasps, which are known for their limited reliability as pollinators (but see ref. [54]). The visitation rate by beetles, which are believed to have a poor affinity for subtle floral phenotypic components[10], was correlated with floral scent and nectar presence. Specifically, species that offered no nectar showed higher visitation rates by beetles, implying a possibility of niche partitioning with the nectar-seeking species in the community. Visitation rate by flies was correlated with aliphatic emissions, but little information is available about volatile preferences by different fly lineages[55]. Butterflies are not major pollinators in Mediterranean scrublands[56], appearing mostly towards the end of the flowering season. Foraging by butterflies has been associated with both visual and olfactory floral stimuli, depending upon the species studied[57]. In addition, adult butterflies can be diverse both in visual physiology and in foraging preferences for different floral traits[57,58]. In our study, tetrachromatic colour as perceived by swallowtail butterflies (viz. saturation and hue component $\varphi_{tet}$)[58], as well as aliphatic emissions, were associated with butterfly visitation rates. The fact that visitation at the level of order is associated with the floral sensory phenotype suggests that, at least in phrygana, the taxonomic orders can represent functional guilds of pollinators.

## Discussion

In this study, we conducted a trait-based analysis to investigate the roles of sensory phenotypic characteristics of insect-pollinated flowers in structuring the p–p network of a natural community, and in shaping plant behaviour in the network. In order to achieve the latter, we (i) removed the effects of phenology by applying a new dynamic data-pooling methodology, (ii) we used a functional approach that incorporated floral abundance into the phenotypic apparency of the species and (iii) we accounted for the effects of phylogenetic affinity between plant species. Thus, what remained to be tested was merely the functional relationship between floral phenotypic traits and species' centrality or specialization. This work represents a step toward the unbiased assessment of multimodal floral stimuli and the role of sensory landscapes in community-wide pollination studies. Despite the limited knowledge of the enormous phenotypic diversity of flowers, and of the innate preferences and sensory systems of

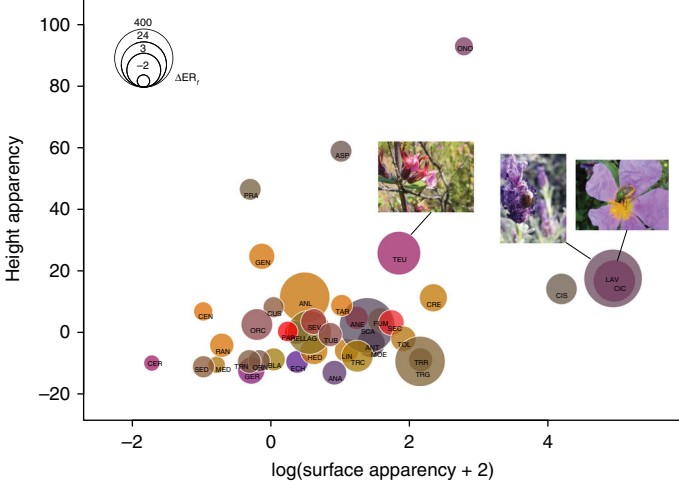

**Fig. 4** Floral apparency in the study community. Apparency values have been calculated by taking into consideration only the co-flowering species (see Methods). Disc radii are scaled versions of the apparency of the total volatile emissions ($\Delta ER_f$). Disc colours approximate the human-perceived colours of the flowers in the RGB system. Flowers of *Teucrium divaricatum* (TEU), *Lavandula stoechas* subsp. *stoechas* (LAV) and *Cistus creticus* (CIC) that exhibit high terpenoid emissions are shown. For the co-flowering species and for the abbreviations of scientific names, see Supplementary Table 6. Photos by A. Kantsa

their visitors, we found significant relationships within a single network, suggesting that our approach could be fruitfully applied to p–p networks elsewhere.

The plant phenonet structure introduced here is centred on the temporally experienced 'flowering lifetime' of each community member with the subset of species that co-flower or are active during that lifetime. Consequently, it is possible to track down every species' behaviour during the active lifetime of every other species in the community. The static pooling would instead only describe species' behaviour during the entire flower season of the area studied or during (biologically) arbitrary time intervals, resulting, inevitably, in inflated specialization or underestimated centrality for the nodes. Different types of mutualistic networks have been shown to exhibit invariant properties, leading to a predictable community structure[5,27]. At the same time, the extent of spatio-temporal scale dependency of mutualistic networks and their analysis is still largely unresolved, although this is crucial to our comprehension of natural complexity in general (e.g. ref. [59]). Thus, any approach that disentangles the drivers that shape network topology and dynamics may be applicable to a variety of ecological systems. Phenological constraints may operate in any interaction network: e.g. fruiting phenology is a known constraint in plant–disperser networks[14]. The phenonet methodology offered here (i) is fully compatible with the temporally dynamic nature of any interaction network, (ii) allows a phenology-unbiased approach and (iii) can be used to more accurately describe the species' roles, specialization and centrality in any network dataset that has been sampled with regular intervals during their periods of activity. In this manner, trait-based analyses may focus on the actual role of phenotypes in shaping interaction patterns, beyond phenology.

Our phenonet results highlight an interesting, and perhaps challenging to interpret, interplay between the ecological and evolutionary trends that shape p–p interactions. For several plant species, ecological roles in the community are plastic in time (Fig. 2) indicating shifting interactions with floral visitors during their phenophases. Given such transitions in visitation dynamics, one might predict a secondary role for floral traits. Instead, our findings show that floral sensory phenotype (visual or chemical) is significantly associated with plant species' behaviour in the phenonet, and that shifting patterns of interactions remain within

the same group of floral visitors (Fig. 2). Under this view, seemingly generalized behaviours of pollinators might include a somewhat cryptic specialization for floral sensory stimuli[29]. Defining functional specialization of species in interaction datasets is already a complicated task[28,29], and given the relationships observed here, the sensory systems of animals as well as the corresponding phenotypes of plants deserve more attention, as the subjects of natural selection. For example, there is evidence that floral colour in several community or regional datasets represents an adaptation to the ancient, well-conserved visual system of the Hymenoptera[28,60,61]. Thus, it is worth examining whether sensory systems that characterize entire clades of insects (e.g. the trichromatic vision of bees or the tetrachromatic vision of butterflies) make them ecologically significant functional guilds, and perhaps attempting to reinvent the 'syndrome' concept[60]. Studies that reveal sensory biases in pollinator perception of flowers, combined with consideration of abiotic (signal efficacy) and biotic (selection from florivores, larcenists and pathogens) effects on floral signal evolution, should lead us to a more nuanced and less typological concept of floral phenotypic 'syndromes'.

Terpenoids are primarily considered as defence compounds[62] and their role in p–p interactions has remained largely unexplored in a community context[63], despite the fact that Schiestl's meta-analysis[64] showed a 90% overlap between the presence of sesquiterpenes as insect pheromones and as floral volatiles. Our results suggest that sesquiterpenes are likely to play multiple ecological roles in plants of the phrygana community, as has been demonstrated for other terpenoid compounds in thyme *Thymus vulgaris* (Lamiaceae) across France[65]. In particular, we suspect that sesquiterpenes function as innate or learned attractants for bees in the phrygana, because (i) bees are by far the most common and diverse pollinators in the region[56], (ii) bee-pollinated plants commonly emit terpenoid scents[55], (iii) sesquiterpene compounds have been shown to attract naïve honeybees[66] and, most importantly, (iv) in our community, sesquiterpene emissions were correlated with floral colorimetric properties as perceived by the trichromatic visual system of the Hymenoptera[28]. Here we found that the quantitative distribution of bee visits among the plants was related to sesquiterpene apparency (Table 2). In addition, the significance of trichromatic floral

**Table 2 Floral traits associated with the quantitative distribution of links in the static network and its sub-networks**

| (Sub-)network | AICsum$_{null}$ | AICsum$_{best}$ | Predictor | Deviance | P-value |
|---|---|---|---|---|---|
| Total | 5166 | 4843 | Trichromatic colour ($r_{tri} \times \theta_{tri}$) | 343.60 | **0.027** |
| | | | %Sesquiterpenes | 352.80 | **0.038** |
| | | | Phenology (start × duration) | 343.60 | **0.030** |
| | | | Surface apparency | 551.10 | **0.004** |
| Coleoptera | 1188 | 1015 | Phenology (start × duration) | 79.94 | **0.009** |
| | | | Symmetry | 90.78 | **0.003** |
| | | | Surface apparency | 158.75 | **0.001** |
| Diptera | 1054 | 1021 | Tetrachromatic colour ($r_{tet} \times \theta_{tet} \times \varphi_{tet}$) | 76.78 | 0.069 |
| | | | Phenology (start × duration) | 88.11 | **0.024** |
| | | | Apparency of total emissions | 90.09 | **0.025** |
| Hymenoptera—Bees | 2035 | 1897 | Phenology (start × duration) | 103.80 | 0.143 |
| | | | Sesquiterpene apparency | 163.60 | **0.011** |
| | | | Height apparency | 192.70 | **0.005** |
| | | | Surface apparency | 206.00 | 0.055 |
| Hymenoptera—Wasps | 353 | 310 | Trichromatic colour ($r_{tri} \times \theta_{tri}$) | 25.87 | 0.128 |
| | | | Surface apparency | 63.17 | **0.012** |
| | | | Phenology (start × duration) | 68.16 | **0.001** |
| Lepidoptera | 462 | 374 | Surface apparency | 63.09 | **0.005** |
| | | | Phenology (start × duration) | 93.36 | **0.001** |

Each subnetwork includes one of the major pollinator groups in the study community. The best multivariate-response generalized linear models (family: negative binomial) according to AIC are presented. The AIC value of the null model (-1) is provided for comparison. Significant P-values (>0.050) are indicated in bold. % Sesquiterpenes: the percentage of sesquiterpene compounds in the total floral emissions. For details on the predictors, see Methods, and the Supplementary Information

colour to the topologically important plants in the network (Table 1) corroborates the fundamental role of bees as important pollinators and selective agents in this habitat type. Accordingly, the PGLS models showed that the variation in bee visitation rates relates to the trichromatic floral hue and increases with sesquiterpene emissions (Table 3). Hence, beyond the signal efficacy hypothesis, it is possible that some groups of bees may have sensory biases to C15 compounds deserving further exploration.

Interestingly, semi-volatile sesquiterpenes have been shown to mediate positive plant−plant associations in natural communities, through diverse mechanisms of conspecific eavesdropping[67], as well as adsorption and re-emission by (heterospecific) neighbouring plants, to defend against herbivores[68]. Given the profusion of sesquiterpenes in Mediterranean scrublands and the association of native plants' emissions with the sensory systems of bees, future research should focus on their broader functional roles across the spectrum of plant–plant interactions, from allelopathy and associational herbivore resistance to facilitation of shared bee pollinators. Specifically, it is worth exploring whether floral scent mediates facilitative interactions in p−p assemblages, as compared with floral morphological traits[28].

Rapid progress has been made towards the development of tools for the functional restoration of pollination networks[7,8], and the identification of plant species with high conservation priority[69]. Given that floral traits relate to plant population vulnerability[70], our approach indicates that sensory-targeted functional restoration and conservation schemes would increase our understanding of and ability to maintain network dynamics and possibly facilitative effects in natural communities. For example, it has been shown that selecting plants for restoration only according to the amount of rewards they offer may undermine facilitative effects and result in undesirable effects due to elevated competition (e.g. ref. [7]). We provide evidence that influential plants in the community may exhibit visual and/or olfactory traits associated with specific pollinator groups, such as bees. Perspectives expand if we consider that floral traits are also involved in antagonistic interactions[71], and the spread of pathogens[72]. Overall, the numerous well-resolved p–p datasets collected across the world are invaluable ecological resources; yet their predictive capacity will remain incomplete as long as floral trait assessments,

focusing on sensory stimuli perceived by floral visitors, remain overlooked or omitted from such studies. Despite the methodological challenges, trait-based analyses of interaction webs improve our understanding of the complexity of natural communities.

## Methods

**Study area.** Sampling was carried out in a coastal thermo-Mediterranean sclerophyllous community (East Mediterranean low scrub, a.k.a. phrygana) in Aghios Stefanos, Lesvos Island, Greece (39°18′.00N, 26° 23′.40E; what three words geocode *reviewers.gladness.hesitantly*). The most dominant flowering plants of the community are *Lavandula stoechas* (French lavender), *Cistus creticus* (pink rock-rose) *Sarcopoterium spinosum* (thorny burnet), and the autumn-flowering *Erica manipuliflora* (autumn heather), with sporadic presence of *Quercus coccifera* (kermes oak), *Pistacia lentiscus* (lentisc) and *Olea europaea* (wild olive tree). The climate is Mediterranean with hot dry summers and mild winters.

**Flower visitation.** Visitation censuses were conducted during the spring flowering periods (April–July) in two successive years (2011–2012). The repetition of observations was important in order to account, to the fullest possible extent, for the above-mentioned high temporal plasticity of species and interactions. We haphazardly established six permanent observation plots with dimensions $3 \times 25 = 75$ m$^2$ in the study area. During both years of sampling, the same observer (AK) visited the study site in 10 day-intervals and recorded all plant–insect interactions during three 15 min diurnal sessions (quarters), distributed from 9 am to 3 pm. During each quarter of an hour, the observer moved at a steady pace within the plot, recording interactions, and collecting insect specimens with a hand net, in case the instant taxonomic identification was not possible at the site. Plant and insect specimens were identified to species or to the lowest taxonomic level possible. An interaction was recorded only when an insect touched the reproductive organs of the flower for more than two seconds. Following a convention in this type of studies[2,8,16,43], these visitors have been used as a proxy for pollinators. In total, 46.7 h were spent observing floral visitation (23.2 h in 2011 and 23.5 h in 2012). To evaluate sampling completeness, we used the approach of Chacoff et al. (based on the Chao 2 estimator)[73], which showed that we detected 73.0% of the total visitor species richness in the community (Supplementary Fig. 3), which is typical in similar studies of p–p networks[73,74].

During every sampling day, all open flower units were counted in the observation plots. As flower units we defined (i) all inflorescences (e.g. compact spikes in *Lavandula*, heads in Asteraceae, dense umbels in Apiaceae), where the distance between individual flowers was so short as to allow small insects to walk on the surface of the inflorescence rather than fly in search of the next source of reward or (ii) the individual flowers that either were solitarily or placed in less compact inflorescences than in (i). Flower density was calculated as flower units per m$^2$, by dividing the total number of units counted in all censuses by the total area observed (250 m$^2$). None of the plants in this community has an exclusively

**Table 3 Floral traits associated with visitation rates by the major pollinator groups in the community**

| Insect group | AICc$_{null}$ | AICc$_{best}$ | Predictor | d.f. | F | P-value | Slope |
|---|---|---|---|---|---|---|---|
| Coleoptera | 53.5 | 45.0 | Nectar | 3,34 | 6.98 | **0.012** | – |
| | | | Al_app × Be_app × Mo_app × Se_app | | 7.92 | **0.004** | + |
| Diptera | 9.0 | 2.4 | Symmetry | 4,33 | 3.38 | 0.075 | – |
| | | | Height apparency | | 3.77 | 0.061 | – |
| | | | %Aliphatics | | 8.34 | **0.007** | – |
| Hymenoptera—Bees | 121.5 | 97.6 | Phenology (start × duration) | 7,30 | 0.00 | 0.928 | – |
| | | | Trichromatic colour ($r_{tri} × \theta_{tri}$) | | 4.35 | **0.045** | – |
| | | | Sesquiterpene apparency | | 4.92 | **0.034** | + |
| | | | %Aliphatics | | 8.92 | **0.006** | + |
| | | | Surface apparency | | 9.34 | **0.005** | + |
| | | | Tetrachromatic saturation ($r_{tet}$) | | 15.29 | **<0.001** | + |
| Hymenoptera—Wasps | −13.9 | −14.3 | Trichromatic colour ($r_{tri} × \theta_{tri}$) | 2,35, | 2.66 | 0.112 | + |
| Lepidoptera | 119.8 | 90.4 | %Al × %Be × %Mo × %Se | 6,31 | 3.83 | 0.059 | – |
| | | | Surface apparency | | 7.53 | **0.010** | – |
| | | | Tetrachromatic hue ($\varphi_{tet}$) | | 16.98 | **0.003** | – |
| | | | Tetrachromatic saturation ($r_{tet}$) | | 10.09 | **<0.001** | – |
| | | | %Aliphatics | | 18.14 | **<0.001** | + |

The best phylogenetically informed generalized least-squares models according to AIC are presented. The AIC value of the null model (-1) is provided for comparison. %Compound class: the percentage of compounds of this class in the total floral emissions. Significant P-values (>0.050) are indicated in bold. Positive or negative relationships are given (slope +: >0, –: <0). For details on the predictors, see Methods and the Supplementary Information
*Al* aliphatics, *Be* benzenoids, *Mo* monoterpenes, *Se* sesquiterpenes, *app* apparency

nocturnal anthesis, and no night surveys were carried out. Specimens are deposited at the *Melissotheque* of the *Aegean*, University of the Aegean, Mytilene, Greece.

**Plant phenonets**. The phenophase of a species represents the time period between the observation days of the first and the last flowering individuals in the community (day numbers of the Julian calendar; Supplementary Tables 1 and 2). Because we have two years of observations, phenophases consist of the average first and the average last days of flowering for every plant of the two years, except for *Geranium robertianum* and *Heliotropium europaeum*, which were only present in the community only in the second year. Accordingly, the phenophase of an insect is defined as the time period between the first and the last day it was recorded in the community irrespective of whether its activity went beyond the time limits of the study. For those insects occurring only in one year, we used the single day numbers of the Julian calendar (Supplementary Table 1).

In order to construct the phenonet for a given plant species, we extracted all other taxa that were present in the community (i.e. 'in flower' for plants, and 'active' for insects) during its flowering phenophase. This sorting required detailed data on the phenology of the community. The aforementioned sampling design, which monitors the visitation network and floral abundance in the community within regular temporal intervals, provided this information. Next, we assembled the realized species–species links within the phenophase (accumulated for the two years of sampling) into 41 new binary networks (Supplementary Table 3). In this way, all node properties attributed to each focal plant correspond to the values obtained from the analysis performed in the own plant's phenonet. This rationale makes use of the dynamic nature of the interaction networks, and assigns corrected roles to the members of the community, by taking into account the phenology of each species and the facts that (i) not all insect species are active during the flowering period of a plant and (ii) not all plant species are flowering during an insect's activity period.

In our study, phenonet data aggregation inevitably assumes that species, even though not observed in one year, may be present in both years. This compromise is necessary in order to acquire the widest possible floral niches of foraging insects. Besides, it distinguishes the phenologically overlapping and thus potentially interacting species within the flowering season.

**Specialization and centrality metrics of species**. We aimed to investigate the role of the visual and chemical floral phenotype in the degree of specialization of the plant species, in the ability of plant species to maintain network cohesion, and to mediate the flow of information within the network. These three questions were addressed by three widely used metrics that describe the behaviour and the importance of the plant species in the p–p community, and are defined below:

1. Normalized Degree (ND), i.e. the number of interacting partners of a species as a proportion of the maximum possible number of interactions in the phenonet. This is the simplest index of centrality for the nodes in a network, and a first measure of their generalized or specialized behaviour[48].
2. Betweenness Centrality (BC). It is a direct measure of the connectivity of a node, proportional to the number of all possible paths connecting all pairs of nodes passing through the focal node. It is computed using the binary unipartite networks of each trophic level. High BC of a species indicates

species whose removal from the network will disrupt the p–p interactions in the entire community or rewire the organization of the paths[47,48].

3. Closeness Centrality (CC). It is considered as the most informative measure of centrality, reflecting the mean distance of a node to other nodes[47]. It is computed using the binary unipartite networks of each trophic level. Higher CC of a species indicates that the species interact with many species with a high ND. It thus reflects how close one plant is to the other co-flowering plants via shared pollinators (pollinator niche of plants).

All the three metrics were calculated in the plant phenonets, with the R package bipartite v. 2.06.

In addition to the above, we calculated the functional role (i.e. network hub, module hub, connector or peripheral) of every plant in the phenonets in which it participates, by computing the among-module connectivity $c$ and the within-module connectivity $z$ (the coordinates in the functional network topography[43]) using the NetCarto software[75]. The same software was used for acquiring the composition of the modules of the static network (Supplementary Table 2). Note that three plants that received no insect visits during the observations, as well as one, which had no co-flowering species (Supplementary Data 1)[41], were not assigned any phenonet node properties. We compared the mean value of each node property calculated for the entire static network vs. the phenonet node properties for all plants (Supplementary Table 4), by employing a paired Wilcoxon signed-rank tests with the function 'wilcox.test' in the R package MASS. Network nestedness was calculated using the function 'nestedness' in the R package bipartite v. 2.06.

**Floral traits**. The floral traits recorded in each one of the 41 insect-pollinated plant species of the community are:

1. Inflorescence scent composition sampled in vivo and in situ, expressed as the relative proportions of each chemical class (viz. aliphatics, benzenoids and phenylpropanoids, monoterpenes, and sesquiterpenes) in the total species blend.
2. Colorimetric properties (viz. key descriptors of saturation and hue) as perceived by either the trichromatic visual system of bees or the potentially higher dimension tetrachromatic visual system of swallowtail butterflies.
3. Presence/absence of nectar.
4. Floral symmetry.
5. Flower height.
6. Frontal surface area of the flower unit.
7. Corolla depth.
8. Flowering phenology.
9. Floral density.

Scent collections were carried out in vivo and in situ with dynamic headspace sampling, except for *Blackstonia perfoliata*, which was sampled in vivo in the lab 1 h after the test plant was collected in the field and transferred to the lab in a pot along with the original soil. All collections were performed once during the peak of the flowering period of a species, on days with clear and calm weather, and at the peak of the pollinators' activity (9 am–1 pm). Scents (Supplementary Data 2)[41] were collected from April to July 2012, except for *Cistus creticus* and *Teucrium*

*divaricatum* from which scents were collected in May–June 2011. On average, we collected four replicate samples of floral headspace for each plant species in the community. During volatile collection sessions, the mean ambient temperature (±SD) was $25.7 \pm 1.4\,°C$, and the mean ambient humidity (±SD) was $55.5 \pm 2.8\%$, measured on the spot. For the dynamic headspace sampling we used a PAS-500 personal air sampler (Supelco, Bellefonte, PA, USA) set at a $200\,mL\,min^{-1}$ flow rate. Only herbivore-free, healthy-looking, fresh inflorescences were selected and enclosed in PET oven roasting bags with a thickness of 12 µm (SANITAS, Sarantis Group, Maroussi, Greece) 10 min prior to sampling. The bagged inflorescences were lightly covered with aluminium foil for shading against sunlight to avoid sweating.

Adsorbent traps contained 10 mg of Porapak® Q (80/100 mesh, Supelco), packed between two plugs of silane-treated glass wool (Supelco) in a borosilicate glass Pasteur pipette (ø 7 mm). The collection period was 90 min for the strongly scented plants (e.g. *Lavandula stoechas*, *Prasium majus*, *Teucrium divaricatum*), and 300 min for the others, as shorter samplings were found insufficient for capturing the volatile profiles of plants with lower emissions. Additionally, during each sampling session, two ambient samples were additionally collected. Samples from green plant parts were collected if possible, in order to detect any compounds from vegetative parts. However, this was not always feasible due to the small size of some plants (e.g. *Anagallis arvensis*, *Sedum confertiflorum*), which would cause extensive tissue damages. In general, we wanted to trap all the compounds emitted naturally by the inflorescence including its green parts (bracts, calyx), and which significantly contribute (especially in Lamiaceae, Apiaceae etc.) to the strong chemosensory environment of pollinators in the Mediterranean shrublands[49].

Immediately after scent collection and yet on the spot, the adsorbent traps were eluted with 300 µL of a 10:1 solution of hexane (puriss. p.a.—Merck, Hohenbrunn, Germany) and acetone (CHROMASOLV® for HPLC—Sigma-Aldrich, Bellefonte, PA, USA). The eluates were stored in a freezer ($-20\,°C$) until chemical analysis. Before analysis, the scent samples were concentrated down to 50 mL with gaseous $N_2$, and 1 ng of toluene (Fluka, Bellefonte, PA, USA) was added as an internal standard in order to estimate emission rates (ER) in toluene equivalents per fresh mass of plant tissue. For the calculation of the ER we used the following formula:

$$\text{Emission rate} = \frac{\left(\sum \frac{\text{peak area of VOC}_i}{\text{peak area of toluene}}\right) \times \text{amount of toluene(ng)}}{\frac{\text{fresh biomass sampled(g)}}{\text{hours of sampling}}} \times \text{concentrated volume } (\mu L)$$

The ER of a compound is expressed in ng (compound, in toluene equivalents) $g^{-1}$ (biomass) $h^{-1}$. For each plant sampled, the average ER for each compound from the different samples was calculated (Supplementary Data 2)[41].

All scent analyses were performed on an Agilent 7890A/5975C GC/MS system (Agilent Technologies, Palo Alto, CA, USA) using splitless injections at 240 °C on a polar GC column (Agilent J&W DB-WAX, length 30 m, ø 0.25 mm, film thickness 0.25 µm) and He as a carrier gas with a flow rate of 1 mL min$^{-1}$. The GC oven was held initially at 40 °C for 3 min and the temperature was increased at 10 °C min$^{-1}$ to 250 °C for 5 min. The two eluents (hexane and acetone) were tested for contaminants using the same method; apart from some other traces, diacetone alcohol (CAS: 123-42-2) was the only abundant contaminant.

We used Agilent MSD Productivity ChemStation software v.E.02.01 (Agilent Technologies) to retrieve the GC/MS data and AMDIS v.2.62 software for peak deconvolution combined with NIST 05 Mass Spectral Library v.2.0d (NIST Mass Spectrometry Data Center, Gaithersburg, MD, USA) to identify VOCs. Kovats retention index was calculated for all the VOCs after analysis of an authentic alkane mix (C10-C40; Sigma-Aldrich) under the above-mentioned chromatographic conditions. Published data on mass spectra and retention times, and authentic standards were additionally used. Whenever possible, we compared VOC retention times and mass spectra to those of authentic standards[28].

In each species' floral scent emissions, we calculated the proportion of each of the four main chemical classes, i.e. aliphatics, benzenoids and phenylpropanoids, monoterpenes and sesquiterpenes. Finally, we estimated the mean normal boiling point (nBP), i.e. the mean boiling point at an atmospheric pressure of 760 mmHg of each plant species' floral volatile blend, as a 'reverse indicator' of a VOC's volatility, following Kantsa et al.[51] (Supplementary Table 6).

To acquire the reflectance spectra of the insect-pollinated flowers of the community we used a portable Jaz spectrometer equipped with a Premium 600 µm reflectance probe (Ocean Optics Inc., Dunedin, FL, USA) with UV transmitting optics. Measurements were taken from the petals or flower units (e.g. entire umbels) and from different individuals; in the case of rare plants we collected as many flower units as available[28]. We measured separately each of the differently coloured areas present on a flower unit (e.g. the yellow tubular and white ligulate florets in *Anthemis* heads), but we included in the analysis only the spectrum (wavelengths: 300–650 nm) of the colour of the largest area[28]. For *Prasium majus* we used spectra extracted from the Floral Reflectance Database (FReD)[76]. Spectral data were processed with the R package pavo v.0.5–1.

We first calculated the polar loci of the plants in the hexagonal colour space of Hymenoptera[77], where the radiant ($r_{tri}$) represents saturation, and the angle ($\theta_{tri}$) represents the hue as perceived by the insect[61]. Apart from the trichromatic vision, we also employed a model representing tetrachromatic information processes (i.e. sensitive to UV, blue, green and red regions of the spectrum) that corresponds to the vision of the swallowtail butterfly *Papilio xuthus* (Papilionidae)[58]. The genus *Papilio* occurs in the study community (*Papilio machaon*, Supplementary

Data 1)[41]. We included this model in order to describe the flower-foraging behaviour of non-bee visitors, since there is reasonable evidence that vision might be trichromatic or tetrachromatic in some cases for different insects, but definitive evidence for specific species is often absent in many cases. The polar expression of the loci in the tetrahedral colour space representing saturation ($r_{tet}$), and the two angles of hue ($\varphi_{tet}$, $\theta_{tet}$)[78] in the swallowtail vision were calculated using function 'tcs' in the the R package pavo v.0.5–1. Reflectance data and the colorimetric properties are included in ref. [28].

For each insect-pollinated plant species in the community, we assessed floral symmetry by distinguishing between actinomorphic and zygomorphic flowers, and corolla depth, by distinguishing shallow (<3 mm) from deep corollas (≥3 mm) (Supplementary Table 6). The latter assessment was based on measuring the length of the actual depth after inserting Drummond microcaps® into flowers[79].

All plants were scored for nectar presence/absence[28] with field observations and by using published data[79].

For flower height, i.e. the average distance of the flower unit from the ground for a given plant species, we took the distance from the ground of the highest and the lowest flower unit per individual (measurement of five individuals for rare taxa, and up to 20 individuals for common ones). Measurements were carried out in the field at the peak of each plant's flowering period using a tape measure (Supplementary Table 6).

For floral surface area, we estimated the area of the flower unit as observed from a frontal view. For actinomorphic or globose flowers and flower units (e.g. *Lagoecia cuminoides*, *Scandix* sp.), we measured the diameter of the circular contour and calculated the frontal area using the formula $A_c = \pi \times radius^2$. For zygomorphic flowers and dense spike-like flower units (e.g. *Lavandula stoechas*) we measured the two perpendicular dimensions of the frontal view and calculated the rectangular area using the formula $A_r = length \times width$. All measurements were performed on 5 (for rare taxa) to 20 flowers per taxon, using a digital calliper (Supplementary Table 6).

**Floral apparency**. We use the term 'apparency' in order to describe the degree to which a plant species in the community has noticeable flowers compared to its co-flowering insect-pollinated species (see analogy in ref. [80]), and we distinguish between visual and chemical flower apparency (Supplementary Fig. 2). For the first one, we used two metrics based on (i) floral height (vertical apparency) and (ii) flower display area (horizontal apparency). We must underline that floral apparency is irrelevant to plant growth form, i.e. species larger in size or perennials are not necessarily more apparent than the other ones in all dimensions examined here (Supplementary Table 6).

The flowers of a plant are apparent unless the co-flowering species have more highly-positioned flowers (compare to ref. [81]); therefore, we calculated flower height apparency as the difference of the median floral height of a plant from the median flower height of the co-flowering species in the community, during its flowering period. Vertical apparency for a focal plant ($\Delta H_f$) is defined as the difference of its floral height from the median floral height of its co-flowering plants in the community:

$$\Delta H_f = H_f - \tilde{H}$$

For the calculation of the median, only the co-flowering plants of the focal species were taken into consideration, in accordance to the phenonet concept (Supplementary Table 3). The values are negative in case the focal plant's flowers are lower-positioned and therefore less apparent than the co-flowering plants.

To estimate horizontal apparency, we first weighted floral surface values for each plant by its mean floral density (flower units per m$^2$) measured in 2011 and 2012. In this way, we obtained the total floral area of each plant species per m$^2$ of flower cover. Floral surface apparency of a focal plant ($\Delta A_f$) was calculated in accordance with height apparency (cf. above), and it represents the difference of the weighted flower area of the focal plant ($A_f$) and the median weighted floral area of the co-flowering plants in the community:

$$\Delta A_f = A_f - \tilde{A}$$

Again, for the calculation of the median, only the co-flowering plants of the focal one were taken into consideration, in accordance with the phenonet concept. Negative values indicate lower horizontal apparency compared with the other co-flowering plants.

Chemical apparency is defined as the difference of the ER of scent per m$^2$ occupied by a given plant ($\Delta ER_f$) from the median ER of the co-flowering plant taxa. The median was used as a measure of central tendency because of the right-skewed distribution of the ER data owing to the large differences of scent among the plants in the community:

$$\Delta ER_f = ER_f - \widetilde{ER}$$

Chemical apparency was calculated for the sum of ER in each plant, as well as for the ER of each one of the four main VOC classes (viz. aliphatics, benzenoids, monoterpenes and sesquiterpenes) (Supplementary Table 5).

Floral apparency is a function of the maximal floral density of each entomophilous plant species in the community. This approach does not intend to

capture the dynamics of floral density exhibited by species during the entire flowering period, but it is used as a measure of the highest possible influence of a given species during its own phenophase. *Heliotropium lasiocarpum* was the last insect-pollinated plant that was in flower alone in the community (Supplementary Table 6). Therefore, for this species, apparency could not be calculated in relation to the co-flowering plants.

**Phylogenies and phylogenetic signal of interactions and traits.** Plant phylogeny was built with the online software Phylomatic v.3 (tree R20120829)[82]. We used 'bladj' algorithm in the software Phylocom v.4.2 (ref. [83]) in order to adjust branch lengths of the phylogeny so as to correspond to evolutionary divergence time between clades, using the most recently updated node ages[84]. The phylogram of the plants of the community is presented in detail in Kantsa et al.[28]. Animal phylogeny (Supplementary Fig. 4) was constructed using the online tool Open Tree of Life[85]. Clade divergence times were retrieved up to the genus level (when impossible, we used the family ages) from the online database TimeTree[86]. Pairwise phylogenetic distance matrices for the two trophic levels were calculated using the function 'cophenetic.phylo' in the R package ape v.3.5.

Phylogenetic dependence between plant and pollinator interacting assemblages is frequently encountered in natural communities, although it is not the rule[23–25]. We used the Procrustean approach to cophylogeny[42], applied with the 'PACo' function in the R package paco v.0.3.2, in order to calculate the phylogenetic congruence of the binary interaction matrix of plants and pollinators in the community. According to this method, plant–pollinator interactions are projected into multivariate space via Principal Coordinates Analysis, undergoing a Procrustean superimposition, where the level of cophylogenetic signal is taken as the global sum of squared residuals ($m^2$) in the best-fit superimposition of the two phylogenies[42]. For each trophic level of the weighted network, we applied separate Mantel tests in the R package vegan v.2.4.0 to calculate the correlation of the interaction dissimilarity matrices of plants and insects (Bray-Curtis index) with their phylogenetic distance matrices[24].

For measuring phylogenetic signal in continuous and in arcsine-transformed proportional independent variables (Supplementary Tables 7 and 8) as well in the visitation rates by the five major pollinator groups (viz. bees, beetles, butterflies, flies and wasps), we used Pagel's $\lambda$, computed with the function 'phyloSignal' in the R package phylosignal v.1.1. Pagel's $\lambda$ is one of the most widely used metrics describing the similarity of the covariances among species to the covariances expected under a Brownian motion model of trait evolution, and it is robust to incompletely resolved phylogenetic trees[87], such as the ones used here; $\lambda$ ranges from 0 (no phylogenetic signal) to 1 (Brownian motion evolution), with intermediate values indicating a trait evolution varying between a "star" phylogeny and Brownian motion.

For binary floral traits (viz. nectar, symmetry and corolla depth), we computed the $D$ statistic of Fritz and Purvis[88], with the function 'phylo.D' in the R package caper v.0.5.2. Values of $D \geq 1$ indicate random distribution of the trait among the tips of the phylogeny, values close to zero indicate evolution by Brownian motion, and values $< -1$ denote extremely phylogenetic clumping.

To test the hypothesis that phylogenetically related plant species are visited primarily by insects of the same group, we used the 'phylo.signal.disc' algorithm developed by Rezende[89], which tests the phylogenetic signal in discrete variables (here, the pollinator group that performed the majority of visits on each species' flowers) by comparing the minimum number of trait-state transitions at each node (accounting for the observed distribution of the trait in the phylogeny in maximal parsimony) with the median of a randomized distribution. A phylogenetic signal is detected when the observed evolutionary transitions are significantly less than the randomized median. The same method was used for detecting phylogenetic signal in module composition. In this case, the discrete variable was the module ID of each species in the community (Supplementary Table 2), tested once for the plant and once for the animal phylogeny.

**Statistical modelling.** We used two modelling approaches in order to explore the role of the above-mentioned elements of floral phenotype into shaping visitation patterns in the community. First, focusing on the plants of the community, we tested which phenotypic traits relate to (i) the visitation rates by the major pollinator groups and to (ii) node properties (BC, CC and ND) of plants. For this, and in order to correct for statistical dependence due to phylogenetic proximity of the plants, we applied phylogenetically informed generalized least-squares models[90] assuming a Brownian motion model of evolution, with the R function 'gls' of package nlme v.3.1.128 that included the phylogenetic correlation matrix of the plants of the community, generated by the R function 'corPagel' in package ape v.4.0. The response variables (ND, BC and CC) as well as the proportion predictor variables (Supplementary Table 8) were arcsine-transformed, whereas height/surface/chemical apparency values were $\log_e + 2$ transformed before analysis. Best-fitting models were selected according to AICc, following a forward selection of variables. Predictors were not strongly correlated with each other (mean absolute value of Pearson's $r = 0.22$, maximum value $= -0.62$).

Second, we tested for the role of floral phenotypic traits (Supplementary Table 8) in shaping the link structure of the overall quantitative (weighted) p–p network. For this, we used a model-based method for compositional data, i.e. Multivariate-response Generalized Linear Models (MGLMs). This approach fits a separate generalized linear model to the visitation matrix of each insect of the network, using a common $n$-dimensional set of explanatory variables and a resampling-based hypothesis testing[91]. The multidimensional response variable is either the entire (static) weighted p–p network or one of its subsets including the insects belonging to the five major groups: Coleoptera, Diptera, Hymenoptera—bees, Hymenoptera—wasps and Lepidoptera. It should be stressed that (i) the residuals of the MGLM model of the p–p network (Manhattan distance matrix) correlate neither with the phylogenetic distance matrix of the plants nor with the insect matrix (Mantel $r_{plants} = -0.13$, $P = 0.915$; $r_{insects} = 0.06$, $P = 0.173$), showing that there is no phylogenetic signal in this analysis and (ii) no phylogenetic signal is detected in the weighted interaction matrix (Mantel tests with Bray-Curtis distance in Supplementary Table 7), allowing us to use a non-phylogenetic modelling approach for this question[92]. For each network, we found the best MGLM model (family: negative binomial) based on the sum of the AIC over all variables following a forward selection of variables. Models were built using the function 'manyglm' in the R package mvabund v.3.12. The statistical significance of the fitted models was assessed with ANOVA (likelihood ratio tests) using 999 bootstrap iterations via PIT-trap residual resampling, a method which shows low rates of type I errors, and in our case yielded the most conservative results. Unless stated otherwise, statistical analyses were performed in R v. 3.3.1 (R Development Core Team; http://www.R-project.org).

**Data availability.** Data on phenophases, floral apparency, floral measurements, network module composition, insect phylogeny and the phylogenetic signal of the tested variables are provided in the Supplementary Information. Plant phylogeny and the flower reflectance spectra are included in ref. [28]. The plant–pollinator network (Supplementary Dataset 1) and the volatile emission rates of the plant species (Supplementary Dataset 2) are available in the online repository Figshare (https://doi.org/10.6084/m9.figshare.5663455).

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

## Acknowledgements

This research was co-financed by EU (European Social Fund) and Greek national funds through the Operational Program "Education and Lifelong Learning" of the National Strategic Reference Framework—Research Funding Program: Heraclitus II (2324-1/WP17/30340). Chemical analyses were carried out at the Laboratory of Water and Air Quality, Department of Environment, University of the Aegean. We are grateful to insect taxonomists: Ante Vujić (Syrphidae), Jelle Devalez and Marino Quaranta (Hymenoptera), and Jos Dils (Bombyliidae). Stefanos Sgardelis and Thanasis Kallimanis provided valuable advice on statistical analysis. A.K. and T.P. thank particularly Themistokles Lekkas, Peter Davies, Maria Aloupi and Olga-Ioanna Kalantzi for support and involvement throughout the research. A.G.D. acknowledges Australian Research Council Discovery Project DP160100161 for support.

## Author contributions

A.K. and T.P. designed the study. A.K. collected the data. A.K., A.G.D., R.A.R. and T.S. analysed the data. The manuscript was written primarily by A.K. with major contributions by A.G.D., R.A.R., T.P., J.M.O. and T.T.
