## [Peer Review File · Nature Communications]

Reviewers' comments:

Reviewer #1 (Remarks to the Author):

I liked this paper. Here the authors present a novel approach to exploring pollination networks using a large dataset including details of pollinator visitation, the relatedness of the plant species and a suite of 'hard' floral traits in a Mediterranean system. What I found most compelling and interesting about this study was the use of phenonets in the network modelling, which led to the illustration that pollination networks are heterogeneous through time with individual species changing network role throughout flowering seasons. To me this finding is what makes this paper, I found this really exciting. The rest, though definitely interesting and novel, is largely correlational and conclusions are based on conjecture. I did find the relationships between given pollinators and groups of floral traits interesting but the best models often included many factors resulting in a string of ideas about why these traits might be grouped together for each type of pollinator. Though this discussion was largely reasonable, I felt it fell short of providing strong evidence-based explanations. In the end the authors did interpret these associations as evidence of cryptic coevolution. I suppose that is not an unreasonable interpretation but at the same time it is not clear if you would find similar groupings in other pollinator networks or not - maybe that is not all that important, but I was not convinced that the observed links between floral traits and insect visitation fundamentally changes my view of coevolved pollination interactions. - I found this aspect of the study "just a start" to understanding if there are generalizable patterns of coevolution between pollinators and floral traits as the authors say themselves.

1) I think you could emphasize the question around your phenonet approach more strongly - bring the question about whether pollination networks are static or plastic more to the front of the introduction. As I said above this is what really stands out to me.

2) One question that I would like to see discussed more in the paper is how the shifting role of species in the network connect with your explanation that these networks are the result of the coevolution of "cryptic specialists" on specific sensory stimuli? On lines 159-164 where you summarise the shifting role of species you point out that species are shifting from generalist to specialist roles and vice versa. This seems slightly discordant with your focus on evolved specialization to specific floral stimuli later in the paper. If species roles change, what happens to their associations with floral traits after those transitions?

3) Following from point 2 - there are two clear sets of key results in your paper - first, that pollination networks are plastic - second, pollinators and plants seem to be connected by sets of floral stimuli. As the moment the link between these parts is not well-articulated and it should be. Your conclusions focus heavily on the floral phenology results and less of the plastic networks structure. I would like to see these parts linked more strongly and to see more emphasis on the plastic network structure throughout - what does this mean for your conclusions about cryptic coevolution?

4) In the conclusions you emphasize the importance of this study as "a first step in the assessment of multimodal floral stimuli" - that is fine but reading the discussion it seemed like you were speculating a lot about what patterns meant. Really, there is nothing in your study that allows you to do anything but speculate about what is driving this diversity or the correlations observed between specific floral stimuli and individual pollinator groups and though I found your explanations plausible, it is going too far based on your presented analyses to conclude that everything depends on these floral stimuli. In general I found your language fair and not overblown but it does seem like a bit of a stretch from the provided analyses to say, as you do on line 307 that: "It appears that evolutionary trends shape interaction diversity, and ultimately, species composition in natural communities."

5) I know you have limited space but I would like you to add a sentence or two to the methods about your actual survey design. Your sampling effort is quite important for understanding how representative your network likely is of what is going on in the system.

6) I am loath to criticize the effort that went into collecting the dataset used in this study as I know that it is just not feasible to survey everywhere all the time. Given that you recorded close to 8000 interactions I also think your survey was sufficient for your purposes here! One thing that does seem to be missing in your survey however, that is potentially important given your focus on olfactory cues, is night pollination. Do you have any evidence that there is little to night pollination going on in this system? or was it just too difficult to survey effectively? It would be good to comment on that in the supplementary material.

7. I do not like how you have handled the calculation of the apparency value for the last species to bloom in your system. If I understand correctly, only this species was blooming at the end of the season and you calculated its apparency in reference to all the other species even though none of them were blooming? This species should be the most apparent species in your network if it is the only resource in this window of time and pollinators are still active. It seems to me your approach will grossly under estimate this species apparency. I would either cut out this species or come up with a way to have the reference state more accurately mirror the reality of negligible apparency.

3. Give better reasons for selecting the three major metrics for describing the centrality and generalization of plants in the phenonets. You just say these are common ones - given that this approach in this context is part of what is great about your study I would really like to see a bit more of a biological rational for choosing these metrics.

Specific comments on phrasing and clarity of information

Overall, I thought this was a really nicely written paper. That said there are a few places where the English is distractingly odd. Probably most importantly, this includes the first paragraph of the whole paper. Take the first sentence (30-31). I would read much better as: "Early in the 20th century, biology adopted network theory in order to investigate complex systems such as food webs."

Lines 37-39- I see what you are trying to do with phrasing the definition of a pollination syndrome this way, but I feel like it then misses that a pollination syndrome is widely viewed as a set of floral features that allow you to predict what the primary pollinators for that species are. Your definition is not wrong but just a bit cryptic as phrased.

Lines 39-43. This sentence does not have logical flow. It does not follow that 1:1 relationships are rare in nature and thus network analyses helped shift the view of pollination to an ecosystem service perspective - this sentence needs to be rewritten, probably split as you make two distinct points with it.

Line 89 -90 - (i.e....) this needs to be changed to: (i.e. the degree of influence a given species has on the network's structure).

Lines 146-147. This seems like a really important result but I do not see at all from what you have said how your analysis tells us that the phenonet approach yields higher ecological generalization than the static network approach. Please explain how you can possible say that.

Lines 165-174. In this paragraph you suddenly become very uncertain in your language and it made me concerned that I was not understanding your results in this section. Why in this results paragraph

alone do you say this “may” do this or that (see line 167 and line 170)? Think about rephrasing throughout this paragraph, I couldn’t tell by the end if you were reporting specific results or discussing possibilities because the results were unclear....

Line 239-257. At this point the writing becomes very obtuse and I had a hard time following the key take home messages – I know it is hard not to be dry when talking about sesquiterpenes but this paragraph definitely needs to be rewritten for readability and for the general audience of this journal. I did not follow (or believe?) your points about why these plants are in a lot of bouquets of the phrygana community. What do you mean by bouquet of the community? the whole community smell? Or do you mean “common in this type of community?” Anyway, this whole paragraph gets away from itself and needs to be rewritten. In addition to adjusting the language, your explanations in this specific paragraph are verging on “just so” stories and go too far.

Line 28 – what do you mean by “plasticity to learn”?

Line 353 – having read your methods, how would you know if the activity went beyond the time limits of the study?

Line 373 – I brought this up before above and I think I see that when you said that phenonets are more generalizable than the static network approach you must mean that the role of specific species are more generalizable – this needs to be made clearer in the text before the methods.

Line 441 – change to “We used two modelling approaches...”

Figure 2 – last sentence of the legend says that plant name abbreviations are in Table S6 but as far as I can tell you have not abbreviated the species names.

Reviewer #2 (Remarks to the Author):

Comments for authors

Disentangling the role of floral sensory diversity in pollination networks

The study by Kantsa et al. tackles an interesting problem, namely how to disentangle the floral sensory diversity in pollination networks. By using a full range of visual and olfactory sensory data of a plant-pollinator interaction network (p-p network) the authors can address a range of interesting questions. These questions are well embedded in two theoretical frameworks: (1) mathematical modelling of (mutualistic) networks, and (2) sensory ecology.

So far, the main goals for p-p network analyses have been to characterize and describe the network structure. Accordingly some of the key questions were: Do pollination/mutualistic networks differ from other networks? What is the level of specialization that we find in these networks? More recently the temporal dynamics of these networks have been investigated. One main goal of these studies were to explore and predict the resilience of the p-p network structure. However, in these analyses species have often been used in a black box approach. In other words the characteristics of species, which are essentially responsible for the formation of network links, were often not considered. Exploring network structure per se has been a powerful approach when it comes to network architecture, network resilience and species dependencies. However, an analysis of network structure per se does not inform us on the mechanisms that are responsible for the formation of species interactions in p-p communities. However, without an understanding of the key mechanisms how p-p networks form discussions on the resilience of p-p networks are without a foundation and somehow artificial.

There have been some few attempts for example to use scent or colour data of flowers and to analyse whether these could explain network structure/formation. However, studies with a more mechanistic approach were often quite limited in their scope. Furthermore, there are several aspects that make the analyses and interpretation of species trait data in the context of network analyses quite difficult. It would go far beyond the scope of a review to go into the details of the lively debates how to interpret scent and colour data as visual and olfactory information from a flower visitor's perspective. Although great advances have been made in the last 20 years in our understanding how to interpret sensory data of flowers in terms of the visual and olfactory system of pollinating insects there are still wide gaps when it comes to the community level. This is largely the result the very successful reductionist approach most scientist have followed to analyse insect responses to visual and olfactory signals. To understand information processing of pollinating insects at the community level in with the goal to identify key signals that impact the interaction between plants and pollinators has been a major challenge.

Kantsa et al. use a novel approach to tackle this problem. The authors introduce the concept of apparency for any given plant species in the community in terms of visual and chemical apparency. This certainly is a critical step in their concept because it makes some assumptions and simplifications. However, I think these assumptions and simplifications are well supported by evidence from field studies. Furthermore, their assumptions lead to testable predictions. For instance, I see some scope in the future to fine tune the apparency model and I could imagine that this could inspire future research. One comment I would like to make: Apparency is here defined from the plant's perspective. I agree with the authors that for several reasons (including practical reasons) the plant perspective was chosen to measuring apparency. Therefore, the assumption was made that plant apparency reflects via the ecological /evolutionary processes apparency from a flower visitor's perspective. These are two different sides of the same medal. The results indeed show that plant apparency plays a key role for explaining network structure in the community/ network. Although this seems plausible it is still an astonishing result because it suggests universal principles of sensory processing among very different pollinator groups. This is also addressed in the discussion. From flower visitor's perspective apparency is the result of innate and/or learned experiences that pollinator individuals integrate over time. Other aspects such as social learning in social bees may add to the complexity of this process. Considering the complexity of scent data, not only when looking at the number of different compounds but also in terms of the multi-functional complexity (defence vs attraction) of floral bouquets it is even more surprising to find such general effects. In the future more sophisticated models might for example also include species specific apparency or dynamics of apparency from a flower visitor perspective.

I was further surprised that sesquiterpene apparency was related to bee visits. This suggests that the role of sesquiterpenes for bee pollinated systems have probably been underestimated. My expectation was that visual signals together with monoterpenes would play the main role for bee systems. The role of sesquiterpenes certainly need to be studied in more detail in the future.

The approach of Kantsa et al. is new in several aspects. By integrating floral sensory data in their models it is possible to gain insights into the underlying mechanisms that may explain interactions between plants and flower visitors. Furthermore, the authors also use a full set of other information sources to thoroughly analyse the data in the context of the phylogenetic relationships, phenology and abundance of interaction partners. I was also positively surprised to see that quantitative data on the flower visitation was used. The study by Kantsa et al. is in several ways remarkable. Firstly, I was amazed by the amount of data that were used to analyse the p-p network. This is the first time that colour and scent data was used in combination to explore the mechanisms for the formation of links in a p-p network. Secondly, the authors were extremely careful to test alternative hypotheses regarding the formation of network links such as phenology and phylogenetic relatedness. I convinced that the spatial/temporal dynamics of species abundance is a key element that may influence for the formation of species interactions. To express the temporal dynamics of the network structure the authors

introduce a new approach for data pooling in bipartite ecological networks (phenonet).

The title is well chosen: There is indeed a great amount of sensory information that might play a role for attracting and recruiting different pollinators. Disentangling the role with respect to different pollinator groups and also considering temporal and phylogenetic effects has always been critical.

The manuscript is very well written. The excellent introduction highlights the main advances that have been made in the last years regarding network analyses of p-p networks and also regarding sensory data and the first attempts to integrate the two fields.

I found that the materials and methods section provides all the information needed to understand how the study was conducted. The methods are well explained. This is particularly important because several new concepts were introduced. The statistical methods are according to what has been used in the field. The supplementary material is very detailed - all steps in the process of analysing the data are made transparent to the reader.

There are several interesting outcomes of the study and I won't list them all: phylogenetically -related plant species showed an overlap in the taxonomy of their pollinators. Furthermore, the authors found that phylogenetically related plants are visited at similar rates by these insect groups. I agree with the authors that these results "...challenge the assumed stochastic nature of linkage rules within the network...and that coevolutionary trends and/or pollinator-specific biases may indeed operate.....".

This is a major contribution because of its interdisciplinary approach and it has a great potential to advance different fields.

Sincerely yours
Andreas Jürgens

Reviewer #3 (Remarks to the Author):

The manuscript presents an interesting and novel way to analyze plant-pollinator networks sampled over two years – the phenonet, which I felt is the most important contribution of the work. In addition, data on the floral sensory stimuli of the plants in the community are presented and the role of these stimuli in shaping the interactions is explored. Here I felt some of the claims of the paper were less convincing and the evidence for "evolutionary trends" and facilitative and competitive relationships was weak and underdeveloped. I am not sure the phylogenetic analyses were done correctly (please see comments regarding Supplemental for more detail). Given methods exist to delve much more thoroughly into this aspect, I am not sure the current analyses are very informative. I would suggest refocusing the paper around the phenonet idea.

Detailed comments:

Abstract

L18: "in unison" is unclear. Do you mean the roles of multiple floral sensory stimuli haven't been addressed in unison? Also, I found the term floral sensory stimuli to be a bit difficult to understand (although it is more clear that just saying "floral sensory diversity" as in the title, as this implies to me that you are referring to the senses of the flowers); perhaps consider floral attractants/advertisements.

L20: I don't think these are opposing ideas, as species can't interact unless they coincide phenologically. So phenological overlap is a prerequisite, not in opposition to the idea that phenotypic traits should underlie interactions.

L21-22: It is unclear to me what these statements have to do with floral sensory stimuli; i.e., how do these goals relate to sensory stimuli?

L25: suggest replacing or defining the term "cohesiveness"

Main text

L30-31: I found this first sentence awkward and unclear. Suggest cutting.

L36: here are you referring to symmetrical specialization being rare? Asymmetrical specialization is not really rare is it?

L59: I had difficulty following the logic behind the transition here.

L61: what do you mean by "pollinator-niche"?

L68-69: shown to relate how/in what way(s)? I think it would be more useful to explain the result rather than just state there was a relationship.

L74: run-on sentence (need semi-colon before thus)

L75: "attachment" is unclear/awkward word choice

L76: how and why does this imply "the hand of evolutionary forces"? Why not a by-product / accident of selection for other reasons?

L77: but floral density isn't static – depends on shape of flowering curve (how floral density changes over the season) and also varies interannually

L85-86: this statement seems overly broad – I think this requires more specificity with regard to what communities you are referring to.

L90-91: I didn't follow this explanation ("simply because this type of data aggregation provides the total number of links between nodes") – how does this explain why studies with temporal resolution on the scale of individual days give static metrics?

L98: Do you mean temporally co-occurring? Although this may be true for direct competition, couldn't species compete indirectly if e.g., earlier flowering species A has a very good year and high floral abundance, this could boost a population of a florivore that feeds on later-flowering species B?

L108: At this point, I was still not sure how you have addressed the dynamics of abundance.

Results and Discussion

L129: I would encourage the complementary analysis to look at whether plant traits showed signal and could explain why closely-related plants shared pollinators. What was the response variable – strength of the interaction or just presence/absence (binary)? Why not look for signal in pollinators, too? Methods exist – Hadfield et al. 2013, Rafferty and Ives 2013.

L133-134: I found this language ("stochastic nature of linkage rules ... and opportunistic nature of interactions") a bit charged and am not sure it is accurate – if retained, I suggest citing literature that espouses these assumptions.

L173-174: how does this indicate keystone species? I think this needs to be explained.

L259: missing word? "including IN other"?

L288: run-on sentence

L307: What is meant by "evolutionary trends"? This is used repeatedly but I felt never really explained.

L308: What is meant by "ecological interdependence" – I didn't think this was measured here (wouldn't this require measures of fitness contributions?)

L314: How have you demonstrated facilitative effects, or even differentiated between facilitative and competitive effects?

Methods

L348: I think you mean day-of-year rather than Julian day, which is a continuous measure of time since the Julian Period.

Supplemental

L38: 46.7 hours of sampling over two years is not very much – do you have evidence that this was sufficient to capture the rarer species (can you do rarefaction analyses or similar to demonstrate this?).

L69: Are there studies you can cite to demonstrate that a single sampling point for floral scents was

adequate?

L156,157,158: The citations numbered 30, 31, 32 do not appear in the list of References (the list ends with citation 29).

L156-157: K is a metric that provides a measure of the strength of signal relative to a model of trait evolution governed by Brownian motion. My understanding is that K should be interpreted with regard to how close the value is to 1 (which corresponds to BM) and that K is sensitive to polytomies in incompletely resolved trees, which seem to be present in your pollinator trees (please see Davies et al. 2012. Incompletely resolved phylogenetic trees inflate estimates of phylogenetic conservatism. Ecology 93:242-247 for a way to correct for this).

Response to Reviewers

Reviewers' comments:

Reviewer #1 (Remarks to the Author):

I liked this paper. Here the authors present a novel approach to exploring pollination networks using a large dataset including details of pollinator visitation, the relatedness of the plant species and a suite of 'hard' floral traits in a Mediterranean system. What I found most compelling and interesting about this study was the use of phenonets in the network modelling, which led to the illustration that pollination networks are heterogeneous through time with individual species changing network role throughout flowering seasons. To me this finding is what makes this paper, I found this really exciting. The rest, though definitely interesting and novel, is largely correlational and conclusions are based on conjecture. I did find the relationships between given pollinators and groups of floral traits interesting but the best models often included many factors resulting in a string of ideas about why these traits might be grouped together for each type of pollinator. Though this discussion was largely reasonable, I felt it fell short of providing strong evidence-based explanations. In the end the authors did interpret these associations as evidence of cryptic coevolution. I suppose that is not an unreasonable interpretation but at the same time it is not clear if you would find similar groupings in other pollinator networks or not - maybe that is not all that important, but I was not convinced that the observed links between floral traits and insect visitation fundamentally changes my view of coevolved pollination interactions. – I found this aspect of the study “just a start” to understanding if there are generalizable patterns of coevolution between pollinators and floral traits as the authors say themselves.

Authors: We warmly thank the Reviewer for their positive reception, and for the most constructive criticism they provided and greatly helped improve the manuscript. Their comments on our reliance on conjecture to explain our results are well taken; they really are “just a start”. However, such criticisms would apply to any correlational study, including nearly all extant studies on plant-pollinator networks, unless they also included unbiased, comprehensive (rather than selective), factorial assays conducted in field settings with live pollinators, independently testing colour, scent and their combinations. We can cite no current study that satisfies these criteria while also providing accurate chemical analysis of floral scent and visual modelling of colour perceptual models for all community members. Junker et al. (2010) have pioneered such studies, but their papers lack chemical analyses and selectively test nodes or hubs (e.g. Junker et al.: 303 links between 35 plant and 164 insect species, of which 18 plant and 10 insect species were tested with choice bioassays). In our study, we identified 401 insect–plant links between 41 plant and 161 insect species. Comprehensively testing all or a subset of these links with behavioural assays would constitute an entirely separate study from the data- and analysis-dense paper we present here. Correlative studies have value when they identify novel hypotheses that can be tested with manipulative experiments. We now identify those next logical experiments in our Discussion.

1) I think you could emphasise the question around your phenonet approach more strongly – bring the question about whether pollination networks are static or plastic more to the front of the introduction. As I said above this is what really stands out to me.

Authors: We thank the Reviewer for this comment. The paragraphs of the Introduction have now been rearranged, however, we would like to highlight that the major scope of this work has been to study the role of floral phenotype in the pollination networks. Phenonets emerged as a necessity in order to tackle the problem of the effects of phenological coupling of species in mutualistic networks, and, in this context, to calculate more accurate specialization/centrality/topological properties of the species. We aimed at a phenology-unbiased approach in order to test merely the associations of floral traits and visitation patterns. The same logic lies behind the apparency approach, and behind the use of phylogenetically-informed statistical models. Thus the phenonet structure is a tool (and apparently one that could have a wide range of applications), which is part of the story. Please bear in mind that not only the plasticity of species' roles, but also the tight correlation between species' behaviour and the floral sensory phenotype are results deriving from the employment of phenonets. We stressed that species roles are plastic, but even so, their behaviour is associated with their sensory phenotype. So, we have retained our discussion about the role of floral sensory diversity in networks, with appropriate disclaimers. Given the additional results we present in the revised version (please see new Fig. 1 and L205-210 & 336-355), it is evident that these structures reveal tight relationships of even the species that shift roles with specific orders of pollinators (e.g. bees). In our opinion, the introduction in its revised version concisely covers the reasoning behind this work; in this context, our coverage of the phenonet approach in the introduction (two paragraphs) which is proportionally equivalent with the extent of the phenonet results and their discussion in the rest of the main text.

2) One question that I would like to see discussed more in the paper is how the shifting role of species in the network connect with your explanation that these networks are the result of the coevolution of "cryptic specialists" on specific sensory stimuli? On lines 159-164 where you summarise the shifting role of species you point out that species are shifting from generalist to specialist roles and vice versa. This seems slightly discordant with your focus on evolved specialization to specific floral stimuli later in the paper. If species roles change, what happens to their associations with floral traits after those transitions?

3) Following from point 2 – there are two clear sets of key results in your paper - first, that pollination networks are plastic – second, pollinators and plants seem to be connected by sets of floral stimuli. At the moment the link between these parts is not well-articulated and it should be. Your conclusions focus heavily on the floral phenology results and less of the plastic networks structure. I would like to see these parts linked more strongly and to see more emphasis on the plastic network structure throughout – what does this mean for your conclusions about cryptic coevolution?

Authors: For points 2) and 3): Thank you very much for these comments, which have been most important for us to address what seemed to be, indeed, results discordant with our interpretation. For this, we explored a bit more carefully the interactions of the species within their phenonets and we found that, although plant "behavioural" roles are shifting (which implies a rearrangement of links among the different nodes) the most species-rich orders that visit each plant remain the same throughout the flowering period (please see new Fig. 1 and L205-210 & 336-355). Given that sensory biases operate at the taxonomic order level (e.g. trichromatic vision of bees, tetrachromatic vision of Diptera, etc.), our findings may support the scenario of cryptic specialization. We added an extra paragraph in order to highlight and to discuss the interplay of the ecological and the evolutionary tension inherent in our results (L333-353).

However, please note that in lines 159-164 of the first version, we summarized the different roles of species in the static network vs. the phenonets to demonstrate that our approach indeed re-evaluates the network metrics of the species. Those lines did not refer to the plasticity of roles that species exhibit during their flowering period.

4) In the conclusions you emphasise the importance of this study as “a first step in the assessment of multimodal floral stimuli” – that is fine but reading the discussion it seemed like you were speculating a lot about what patterns meant. Really, there is nothing in your study that allows you to do anything but speculate about what is driving this diversity or the correlations observed between specific floral stimuli and individual pollinator groups and though I found your explanations plausible, it is going too far based on your presented analyses to conclude that everything depends on these floral stimuli. In general I found your language fair and not over blown but it does seem like a bit of a stretch from the provided analyses to say, as you do on line 307 that: “It appears that evolutionary trends shape interaction diversity, and ultimately, species composition in natural communities.”

Authors: Thank you for this criticism. Please note that we are not claiming that everything depends on floral stimuli. What we have shown is that when a comprehensive set of floral sensory stimuli (parameterized in a human-unbiased way) are included in our model along with floral abundance and physical apparency, the model tells us that sensory traits can be highly correlated with plant behaviour in the network. Importantly, these conclusions were produced after removing the effects of phenology, and after accounting for the phylogenetic similarity of the plant species. These results have, in our opinion, useful implications in conservation of p–p communities. This information has never been provided before in p–p network studies, although there are excellent papers in the recent literature that have studied the role of specific floral stimuli in the p–p networks (Junker et al. 2010, Junker et al. 2013, Renoult et al. 2015, Larue et al. 2016)

Unfortunately, until all the involved sensory systems are adequately explored, and until behavioural experiments are conducted in order to test the neurophysiological or psychophysical responses of different pollinators to the sensory floral stimuli, our inferences represent future research questions regarding the functional importance of these traits. Methodologically, we employed the latest tools that sensory ecology of pollination provides, and we are confident about that.

The line you are citing at the end of the comment has been removed because the first paragraph of the Conclusions changed in large part (please see revised version L357-369). In the revised version, we tried to carefully address coevolution of p–p interactions. Furthermore, we added phylogenetic analyses (see L133-146, L666-676), which show that coevolution of insects and plants is strongly related to the entire binary p–p network, as well as we find that the modular structure is associated with insect phylogeny. Even plants' Betweenness Centrality shows a phylogenetic signal (new Table S9). There is definitely a very interesting, and perhaps still difficult to interpret, interplay between ecological and evolutionary specialization at a community-level. But we find that taxonomic orders of insects are important for shaping interactions (L154-163), their visitation rates correlate with sensory floral traits (Tables 2-3), and we know that sensory systems (e.g. vision) often vary at the level of order (e.g. Chittka and Briscoe 2001, Dyer et al. 2012). We summarized all these points, trying to use an even fairer language than before, in the lines 336-355.

5) I know you have limited space but I would like you to add a sentence or two to the methods about your actual survey design. Your sampling effort is quite important for understanding how representative your network likely is of what is going on in the system.

Authors: We moved this part of methodology from the Suppl. Material to the main text. Please see the revised Methods. (L365-424).

6) I am loath to criticize the effort that went into collecting the dataset used in this study as I know that it is just not feasible to survey everywhere all the time. Given that you recorded close to 8000 interactions I also think your survey was sufficient for your purposes here! One thing that does seem to be missing in your survey however, that is potentially important given your focus on olfactory cues, is night pollination. Do you have any evidence that there is little night pollination going on in this system? or was it just too difficult to survey effectively? It would be good to comment on that in the supplementary material.

Authors: Based on earlier studies carried out on Lesvos Island (Nielsen et al. 2011, Lázaro et al. 2016a, Lázaro et al. 2016b), we know that none of our plants exhibited exclusively nocturnal pollination, so we set the context of study as the time of activity of diurnal pollinators. However, if nocturnal visitation occurs (e.g. by noctuid moth species), we did not score it. We added one sentence to make this clear (please see revised version, L421-422).

7. I do not like how you have handled the calculation of the apparency value for the last species to bloom in your system. If I understand correctly, only this species was blooming at the end of the season and you calculated its apparency in reference to all the other species even though none of them were blooming? This species should be the most apparent species in your network if it is the only resource in this window of time and pollinators are still active. It seems to me your approach will grossly underestimate this species apparency. I would either cut out this species or come up with a way to have the reference state more accurately mirror the reality of negligible apparency.

Authors: The Reviewer is right. According to their suggestion, we re-ran the multivariate-response generalized linear and the PGLS models without *Heliotropium lasiocarpum*.

3. Give better reasons for selecting the three major metrics for describing the centrality and generalization of plants in the phenonets. You just say these are common ones - given that this approach in this context is part of what is great about your study I would really like to see a bit more of a biological rationale for choosing these metrics.

Authors: We modified the text in order to highlight the importance of the three selected metrics. Please note that the ecological significance of each one was already mentioned in each metric's description. Please see revised version, L452-471.

Specific comments on phrasing and clarity of information

Overall, I thought this was a really nicely written paper. That said there are a few places where the English is distractingly odd. Probably most importantly, this includes the first paragraph of the whole paper. Take the first sentence (30-31). I would read much better as: “Early in the 20th century, biology adopted network theory in order to investigate complex systems such as food webs.”

Authors: Changed as suggested. We also revisited and simplified several parts of the text. We hope it is better now.

Lines 37-39- I see what you are trying to do with phrasing the definition of a pollination syndrome this way, but I feel like it then misses that a pollination syndrome is widely viewed as a set of floral features that allow you to predict what the primary pollinators for that species are. Your definition is not wrong but just a bit cryptic as phrased.

Authors: Thank you for this comment. Ollerton et al. (2009) define pollination syndromes as “*suites of convergent floral traits hypothesized to adapt distantly related angiosperm species to particular types of pollen vectors*” (Ollerton et al. 2009). Fenster et al. (2004) state that “*pollination syndromes provide great utility in understanding the mechanisms of floral diversification*” (Fenster et al. 2004). Given these, we reckon that the definition of floral syndromes includes the importance of pollen vectors for floral evolution. Thus, we chose not to change this sentence.

Lines 39-43. This sentence does not have logical flow. It does not follow that 1:1 relationships are rare in nature and thus network analyses helped shift the view of pollination to an ecosystem service perspective – this sentence needs to be rewritten, probably split as you make two distinct points with it.

Authors: The Reviewer is right. This sentence has been modified. Please see revised version, L44-46.

Line 89 -90 – (i.e....) this needs to be changed to: (i.e. the degree of influence a given species has on the network’s structure).

Authors: Changed as suggested.

Lines 146-147. This seems like a really important result but I do not see at all from what you have said how your analysis tells us that the phenonet approach yields higher ecological generalization than the static network approach. Please explain how you can possibly say that.

Authors: Thank you for observing this. This paragraph along with the previous one needed rearrangement. Regarding the increased generalization of the plants in the phenonets, we are referring to the results presented in Fig. 2 and Table S4. We now think it is clearer. Please see revised version, L182-198.

Lines 165-174. In this paragraph you suddenly become very uncertain in your language and it made me concerned that I was not understanding your results in this section. Why in this results paragraph alone do you say this “may” do this or that (see line 167 and line 170)? Think about rephrasing throughout this paragraph, I couldn’t tell by the end if you were reporting specific results or discussing possibilities because the results were unclear....

Authors: Thank you. We did not intend to cause doubt or uncertainty, rather than possibility. However, this paragraph has now been changed, as new results have been added. Please see the revised version, L199-216.

Line 239-257. At this point the writing becomes very obtuse and I had a hard time following the key take home messages – I know it is hard not to be dry when talking about sesquiterpenes but this paragraph definitely needs to be rewritten for readability and for the general audience of this journal. I did not follow (or believe?) your points about why these plants are in a lot of bouquets of the phrygana community. What do you mean by bouquet of the community? the whole community smell? Or do you mean “common in this type of community?” Anyway, this whole paragraph gets away from itself and needs to be rewritten. In addition to adjusting the language, your explanations in this specific paragraph are verging on “just so” stories and go too far.

Authors: Thank you. The paragraph was indeed difficult to understand. We re-wrote it, trying to explain more clearly our view on the functional role of sesquiterpenes in the community, and to suggest future research questions, based on our results and on the current literature (revised version, L269-310).

Line 28 – what do you mean by “plasticity to learn”?

Authors: The sentence has been changed. Please see revised version, L326-328.

Line 353 – having read your methods, how would you know if the activity went beyond the time limits of the study?

Authors: It is true that we do not know the exact duration of an insect’s activity. This would have required exhaustive sampling, beyond the scope of this work. Nevertheless, the frequency of our sampling makes us confident that the phenology recorded is a good proxy for the true phenology of pollinators in the study community. Besides, because in the Mediterranean systems the majority of the pollinators consist of solitary bees (Petanidou and Ellis 1993, Michener 2007), we expect that the biggest fraction of them have a relatively narrow adult life span, i.e. up to a few weeks long. Of course, there are some multivoltine species (i.e. those

having >1 broods of offspring per year), e.g. of the family Halictidae, having much longer life spans, e.g. 2-3 months or longer. We did not follow the development of the system outside the period of the high peak of flowering, as it was out of the scope of our work.

Line 373 – I brought this up before above and I think I see that when you said that phenonets are more generalizable than the static network approach you must mean that the role of specific species are more generalizable – this needs to be made clearer in the text before the methods.

Authors: This is clarified in the text before the Methods (revised version, L182-183), and we adapted the relevant Methods subsection's title (L451).

Line 441 – change to “We used two modelling approaches...”

Authors: OK.

Figure 2 – last sentence of the legend says that plant name abbreviations are in Table S6 but as far as I can tell you have not abbreviated the species names.

Authors: The names of the phenonets (top row shaded in black) were abbreviated species names. We now modified the caption to make it clearer.

Reviewer #2 (Remarks to the Author):

Comments for authors

Disentangling the role of floral sensory diversity in pollination networks

The study by Kantsa et al. tackles an interesting problem, namely how to disentangle the floral sensory diversity in pollination networks. By using a full range of visual and olfactory sensory data of a plant-pollinator interaction network (p-p network) the authors can address a range of interesting questions. These questions are well embedded in two theoretical frameworks: (1) mathematical modelling of (mutualistic) networks, and (2) sensory ecology.

So far, the main goals for p-p network analyses have been to characterize and describe the network structure. Accordingly some of the key questions were: Do pollination/mutualistic networks differ from other networks? What is the level of specialization that we find in these networks? More recently the temporal dynamics of these networks have been investigated. One main goal of these studies were to explore and predict the resilience of the p-p network structure. However, in these analyses species have often been used in a black box approach. In other words the characteristics of species, which are essentially

responsible for the formation of network links, where often not considered. Exploring network structure per se has been a powerful approach when it comes to network architecture, network resilience and species dependencies. However, an analysis of network structure per se does not inform us on the mechanisms that are responsible for the formation of species interactions in p-p communities. However, without an understanding of the key mechanisms how p-p networks form, discussions on the resilience of p-p networks are without a foundation and somehow artificial.

There have been some few attempts for example to use scent or colour data of flowers and to analyse whether these could explain network structure/formation. However, studies with a more mechanistic approach were often quite limited in their scope. Furthermore, there are several aspects that make the analyses and interpretation of species trait data in the context of network analyses quite difficult. It would go far beyond the scope of a review to go into the details of the lively debates how to interpret scent and colour data as visual and olfactory information from a flower visitor's perspective. Although great advances have been made in the last 20 years in our understanding how to interpret sensory data of flowers in terms of the visual and olfactory system of pollinating insects, there are still wide gaps when it comes to the community level. This is largely the result the very successful reductionist approach most scientist have followed to analyse insect responses to visual and olfactory signals. To understand information processing of pollinating insects at the community level in with the goal to identify key signals that impact the interaction between plants and pollinators has been a major challenge.

Kantsa et al. use a novel approach to tackle this problem. The authors introduce the concept of apparency for any given plant species in the community in terms of visual and chemical apparency. This certainly is a critical step in their concept because it makes some assumptions and simplifications. However, I think these assumptions and simplifications are well supported by evidence from field studies. Furthermore, their assumptions lead to testable predictions. For instance, I see some scope in the future to fine tune the apparency model and I could imagine that this could inspire future research. One comment I would like to make: Apparency is here defined from the plant's perspective. I agree with the authors that for several reasons (including practical reasons) the plant perspective was chosen to measuring apparency. Therefore, the assumption was made that plant apparency reflects via the ecological /evolutionary processes apparency from a flower visitor's perspective.

These are two different sides of the same coin. The results indeed show that plant apparency plays a key role for explaining network structure in the community/ network. Although this seems plausible it is still an astonishing result because it suggests universal principles of sensory processing among very different pollinator groups. This is also addressed in the discussion. From flower visitor's perspective apparency is the result of innate and/or learned experiences that pollinator individuals integrate over time. Other aspects such as social learning in social bees may add to the complexity of this process. Considering the complexity of scent data, not only when looking at the number of different compounds but also in terms of the multi-functional complexity (defence vs attraction) of floral bouquets it is even more surprising to find such general effects. In the future more sophisticated models might for example also include species specific apparency or dynamics of apparency from a flower visitor perspective.

I was further surprised that sesquiterpene apparency was related to bee visits. This suggests that the role of sesquiterpenes for bee pollinated systems have probably been underestimated. My expectation was that visual signals together with monoterpenes would play the main role for bee systems. The role of sesquiterpenes certainly need to be studied in more detail in the future.

The approach of Kantsa et al. is new in several aspects. By integrating floral sensory data in their models it is possible to gain insights into the underlying mechanisms that may explain interactions between plants and flower visitors. Furthermore, the authors also use a full set of other information sources to thoroughly analyse the data in the context of the phylogenetic relationships, phenology and abundance of interaction partners. I was also positively surprised to see that quantitative data on the flower visitation was used. The study by Kantsa et al. is in several ways remarkable. Firstly, I was amazed by the amount of data that were used to analyse the p-p network. This is the first time that colour and scent data was used in combination to explore the mechanisms for the formation of links in a p-p network. Secondly, the authors were extremely careful to test alternative hypotheses regarding the formation of network links such as phenology and phylogenetic relatedness. I am convinced that the spatial/temporal dynamics of species abundance is a key element that may influence for the formation of species interactions. To express the temporal dynamics of the network structure, the authors introduce a new approach for data pooling in bipartite ecological networks (phenonet).

The title is well chosen: There is indeed a great amount of sensory information that might play a role for attracting and recruiting different pollinators. Disentangling the role with respect to different pollinator groups and also considering temporal and phylogenetic effects has always been critical.

The manuscript is very well written. The excellent introduction highlights the main advances that have been made in the last years regarding network analyses of p-p networks and also regarding sensory data and the first attempts to integrate the two fields.

I found that the materials and methods section provides all the information needed to understand how the study was conducted. The methods are well explained. This is particularly important because several new concepts were introduced. The statistical methods are according to what has been used in the field. The supplementary material is very detailed - all steps in the process of analysing the data are made transparent to the reader.

There are several interesting outcomes of the study and I won't list them all: phylogenetically-related plant species showed an overlap in the taxonomy of their pollinators. Furthermore, the authors found that phylogenetically related plants are visited at similar rates by these insect groups. I agree with the authors that these results "...challenge the assumed stochastic nature of linkage rules within the network...and that coevolutionary trends and/or pollinator-specific biases may indeed operate.....".

This is a major contribution because of its interdisciplinary approach and it has a great potential to advance different fields.

Sincerely yours

Andreas Jürgens

Authors: We deeply thank Dr Jürgens for appreciating the amount of work performed, and for so kindly acknowledging the scope and necessity of our study.

Reviewer #3 (Remarks to the Author):

The manuscript presents an interesting and novel way to analyze plant-pollinator networks sampled over two years – the phenonet, which I felt is the most important contribution of the work. In addition, data on the floral sensory stimuli of the plants in the community are presented and the role of these stimuli in shaping the interactions is explored. Here I felt some of the claims of the paper were less convincing and the evidence for “evolutionary trends” and facilitative and competitive relationships was weak and underdeveloped. I am not sure the phylogenetic analyses were done correctly (please see comments regarding Supplemental for more detail). Given methods exist to delve much more thoroughly into this aspect, I am not sure the current analyses are very informative. I would suggest refocusing the paper around the phenonet idea.

Authors: We warmly thank Reviewer #3 for their most constructive criticism that helped to improve the manuscript. The phenonet concept constitutes a tool for accounting for the effect of phenological uncoupling to estimating species properties in the p–p network. We are glad to see that it is found to be meaningful by the Reviewers, yet we would like to highlight that it is only part of the story we want to present to the reader.

The Reviewer’s comments triggered us to include more phylogenetic analyses in the manuscript, as well as replacing a metric of phylogenetic signal due to potential doubts on its performance, although we disagree with their view that the analyses in the first manuscript were not done correctly. The additional analyses helped to better interpret our results.

Also, please note that we never intended to discuss in detail facilitation or competition, as these processes would have required additional experiments and, after all, they were not part of the scope of this work. Instead, they were mentioned only to discuss our findings and to describe future implications in the conservation of p–p communities.

We provide more details in the responses to the specific comments below.

Detailed comments:

Abstract

L18: “in unison” is unclear. Do you mean the roles of multiple floral sensory stimuli haven’t been addressed in unison? Also, I found the term floral sensory stimuli to be a bit difficult to understand (although it is more clear that just saying “floral sensory diversity” as in the title, as this implies to me that you are referring to the senses of the flowers); perhaps consider floral attractants/advertisements.

Authors: According to the Reviewer’s suggestion, we changed “assessed” with “addressed”. A stimulus is a function-neutral way to refer to phenotypic characteristics with potential sensory properties. The terms “signal” and “cue” have strict meanings in ethology (Searcy and Nowicki 2005, Saleh et al. 2007), and one cannot use the term “attractant” without clear demonstration via bioassay (e.g. Ayasse et al. 2003). “Sensory stimuli” implies the potential for these functions and is more concise than repeating “floral colour, shape and scent”. In the revised version, we use the term “attractant” only for discussing possible explanations of our results (L290, 319), and not to address the floral traits that we have measured. Following the Reviewer’s concern, we reconsidered the title of the manuscript and modified it into: “**Disentangling the role of floral sensory stimuli in pollination networks**”.

L20: I don't think these are opposing ideas, as species can't interact unless they coincide phenologically. So phenological overlap is a prerequisite, not in opposition to the idea that phenotypic traits should underlie interactions.

Authors: We agree, but there has been a tendency to overlook phenotype compared with phenological trends, which has had the effect (intentionally or not) of omitting sensory ecology and foraging rules from the plant-pollinator network literature (see literature cited in the Introduction). We only used "whereas" because we wanted to emphasize that the majority of studies focus on phenology and floral density, largely overlooking plant functional phenotypic traits. However, the sentence has now been changed (revised version, L17-19).

L21-22: It is unclear to me what these statements have to do with floral sensory stimuli; i.e., how do these goals relate to sensory stimuli?

Authors: We have re-written this statement for clarity: *'Despite progress in understanding pollination network structure, the functional roles of floral sensory stimuli (visual, olfactory) have never been addressed comprehensively in a community-context, even though such traits are known to mediate plant-pollinator interactions'*. (revised version, L17-19).

L25: suggest replacing or defining the term "cohesiveness"

Authors: According to the Reviewer's suggestion, we removed the term from the Abstract, in order to make it more accessible to general readership. After a more careful examination, we replaced "cohesiveness" with "cohesion" throughout the text, which is the grammatically correct term.

Main text

L30-31: I found this first sentence awkward and unclear. Suggest cutting.

Authors: We adapted the sentence in order to make it clearer. Please see revised version, L30-31.

L36: here are you referring to symmetrical specialization being rare? Asymmetrical specialization is not really rare is it?

Authors: We do not refer to asymmetrical specialization, so we clarified this to avoid confusion. Please see revised version, L42.

L59: I had difficulty following the logic behind the transition here.

Authors: Thank you for this comment. The way it was written, it could have been easily interpreted that we would test whether these trait combinations really do (behaviourally) attract different pollinators. But this was not our scope. So, we changed these few sentences: *'Recently, our study revealed a phenotypic integration between floral color (as perceived by pollinators) and scent at a community level among the flowering plants; this finding suggests a coordinated adaptation of plants to the sensory systems of pollinating insects. We build upon this study by asking whether floral phenotypes that match visitors' physical and sensory biases in a community-context represent evolutionary vestigial traits or relics with no extant function, or alternatively, whether they are correlated with the realized pollinator-niches of the plants'* (revised version, L61-67). We hope that the transition is clearer now.

L61: what do you mean by "pollinator-niche"?

Authors: We used this term to address the suite of pollinators that visit a plant species (see also Johnson and Raguso 2016).

L68-69: shown to relate how/in what way(s)? I think it would be more useful to explain the result rather than just state there was a relationship.

Authors: Small adjustments have been made in order to present the result of Renoult et al. (2015) in a clearer and more meaningful way (revised version, L74-76).

L74: run-on sentence (need semi-colon before thus)

Authors: The sentence has been changed (revised version, L76-80).

L75: "attachment" is unclear/awkward word choice

Authors: "Preferential attachment" is an established term describing the tendency of new species to link to species that already have many links in the network (Barabasi and Albert 1999, Olesen et al. 2008). However, in the new version of the Introduction, we no longer include this term.

L76: how and why does this imply "the hand of evolutionary forces"? Why not a by-product / accident of selection for other reasons?

Authors: That exactly was the question: *'(i) does preferential attachment between plant and pollinator nodes in the network³⁰ have a sensory basis, implying the hand of evolutionary forces, and (ii) is this behavior simply a function of floral density[...]?'* (first version, L75-77).

The “hand of evolution” in this hypothesis (i.e. that preferential attachment could be explained by flower–pollinator coevolution) is implied because floral stimuli can be co-evolved with the sensory systems and innate preferences of pollinators (e.g. Lunau 2004, Schiestl 2010, Schiestl and Dotterl 2012). Our recent results in the same community (Kantsa et al. 2017) show that the visual and chemical floral phenotype of the plants are integrated, and, in this context, specific chemical properties are correlated with the colorimetric properties of the visual systems of bees and of swallowtail butterflies.

However, please note that this section has been largely changed (revised version, L76-80).

L77: but floral density isn't static – depends on shape of flowering curve (how floral density changes over the season) and also varies interannually

Authors: It is true that floral density is not static. We primarily intended to address the dynamics of interactions and of the network's structure. The indices of floral apparency that we introduce reflect the highest floral density of each plant species and the maximal influence that the floral phenotype of one species may exert in the community. We added a few explanatory lines in the Methods section (revised version, L652-655).

L85-86: this statement seems overly broad – I think this requires more specificity with regard to what communities you are referring to.

Authors: The references we cited in this sentence represent studies of two very different communities, both biogeographically, and ecologically: a Mediterranean scrubland in Greece (Petanidou et al. 2008), and a boreal heathland in Greenland (Olesen et al. 2008). This is why we had made this statement seem broad. However, in an attempt to use fairer language, we modified the text in these lines: *'The resulting inflation can be considerable, given that most species in many communities, including the Mediterranean ones, tend to have short flowering phenophases'* (revised version, L88-90).

L90-91: I didn't follow this explanation (“simply because this type of data aggregation provides the total number of links between nodes”) – how does this explain why studies with temporal resolution on the scale of individual days give static metrics?

Authors: In order to calculate specialization, we need to take into consideration the number of the realized interactions against the number of the possible ones. In the p–p network studies published so far, specialization of the nodes (species) was estimated by employing metrics that take into consideration the entire network matrix (the total number of species of the interacting trophic level) in order to calculate node (species) properties.

For example, for calculating Normalised Degree ($ND = \text{no. realized links} / \text{no. possible links}$) of a plant species with the traditional static approach, one would necessarily use as denominator the total number of insect species in the community. But this is not realistic: not all species co-occur temporally with every other one in the community. The static approach is predicted (and proved in our manuscript; Fig. 2), to produce lower

values of ND for the plants. Slicing a network on a weekly basis is arbitrary and does not necessarily make biological sense for calculating species' specialization, centrality metrics or the topological roles, because species interact in between these intervals.

The plant phenonet structure differs because it is centred on the temporally experienced "floral lifetime" of each community member with the subset of species (plants and insects) that co-flower during that lifetime. In contrast, slicing a static network into smaller and smaller time slices (as done finely in Rasmussen et al. 2013) without changing the focus would continue to misrepresent possible interactions when calculating species' properties (specialization, centrality etc.). With the phenonets, it is possible to track down every species' behaviour during the lifetime of every other species in the community. The static pooling would only describe species' behaviour during the entire flower season of the area studied or during (biologically) arbitrary time intervals, resulting, inevitably, in inflated specialization or underestimated centrality for the nodes.

L98: Do you mean temporally co-occurring? Although this may be true for direct competition, couldn't species compete indirectly if e.g., earlier flowering species A has a very good year and high floral abundance, this could boost a population of a florivore that feeds on later-flowering species B?

Authors: Yes, we mean temporally co-occurring and we added this in the text (L102 in the revised version). However, please note that the suggested scenario could as well "boost" the population of pollinators or encourage migratory ones to remain in the population in a facilitative sense, as described classically by Waser and Real (1979). The Reviewer is correct in referring to "sequential mutualism", a phenomenon that may be true either for migratory pollinators, as well as for species that share pollinators, although they do not co-flower in the same community (Waser and Real 1979). However, we opt to consider what was actually recorded rather than imagining the potential relationships which might be innumerable, given the generalist species within the community.

L108: At this point, I was still not sure how you have addressed the dynamics of abundance.

Authors: Please see our response to the comment for L77.

Results and Discussion

L129: I would encourage the complementary analysis to look at whether plant traits showed signal and could explain why closely-related plants shared pollinators. What was the response variable – strength of the interaction or just presence/absence (binary)? Why not look for signal in pollinators, too? Methods exist – Hadfield et al. 2013, Rafferty and Ives 2013.

Authors: In this paragraph, the trait tested for phylogenetic signal was the predominant pollinator group (i.e. the group that performs the majority of visits on the flowers of each species over the two-year observations), which was treated as a categorical variable (revised version Fig. S1a; bees, wasps, beetles, butterflies, and flies) produced by taking into account the quantitative data p-p on interactions.

Following the Reviewer's suggestion on expanding the analyses related to the phylogenetic structure of the community we:

- 1) Added an extra supplementary table including the phylogenetic signal of all the independent variables used in our models (Table S8).
- 2) Constructed the phylogeny of the pollinator community (see L666-673. Fig. S5).
- 3) Calculated the cophylogenetic signal of the entire network, test if the coupled phylogenetic history of the two trophic levels is associated with network's structure, using the Procrustean Approach to Cophylogeny (PACo) developed by Balbuena et al. (Balbuena et al. 2013), applied with the R package *paco* (Hutchinson et al. 2017a, Hutchinson et al. 2017b). According to this method, p-p interactions are projected into multivariate space via Principal Coordinates Analysis, undergoing a Procrustean superimposition, where the level of cophylogenetic signal is taken as the global sum of squared residuals in the best-fit superimposition of the two phylogenies (revised version, L670-673). This approach yielded similar results with ParaFit (Legendre et al. 2002), which is also used for calculating coevolution.
- 4) Measured the modularity of the network according to Olesen et al. (Olesen et al. 2007) and tested the module composition for phylogenetic signal.

These additional analyses provided very interesting results showing that, indeed, the interactions are shaped by plant-insect coevolution, and that insect phylogeny is highly associated with the distribution of plant and animal species into modules. All these results are discussed in detail in the revised version (L133-163).

Furthermore, the new analyses are indeed complementary to the phylogenetic generalized least squares (PGLS) regression models we had already presented in the first version, a method which is widely used for more the last 10 years in multispecies datasets (Paradis 2012, Jürgens et al. 2013, Anacker et al. 2014, Gomez et al. 2014, Swenson 2014, Klempay et al. 2016, Kantsa et al. 2017).

Following the Reviewer's suggestion, we also tried to implement the PLMMs by Rafferty and Ives (2013) using the R package *pez* (Pearse et al. 2015). The approach seemed hopeful because it implemented both participant phylogenies (plants and animals) with the potential to test each floral trait upon the network structure. Unfortunately, this approach has its own shortcomings, which ultimately did not lead us to the resolution recommended by the reviewer. These include but are not limited to:

- 1) The function 'pglm' that runs the models cannot provide Akaike Information Criteria for each GLM tested, such that we could not compare models nor gauge the effects of our traits of interest
- 2) The size of our matrix appears to be prohibitive for the kinds of iterative computational effort needed, (e.g. handle multiple predictors) again, without the opportunity of ultimately comparing models
- 3) There is limited feedback from other studies using the Rafferty and Ives approach, so we could not trouble shoot based on the experiences of others.

Finally, we also considered the approach outlined by Hadfield et al. Hadfield et al. (2014) but at present the appropriate code is not available.

Overall, in our work, we used tested methods (i) to detect phylogenetic signal in floral traits, (ii) to detect the cophylogenetic signal of the network, and (iii) to conduct phylogenetic regression. The only non-phylogenetic method we use are the MGLMs, which we were allowed to use because the quantitative interaction matrix of the network does not show significant correlation neither with the plant nor with the animal phylogenetic distance matrix (Kamilar and Cooper 2013) (Table S8, L721-727).

L133-134: I found this language (“stochastic nature of linkage rules ... and opportunistic nature of interactions”) a bit charged and am not sure it is accurate – if retained, I suggest citing literature that espouses these assumptions.

Authors: These terms are frequently used by several authors describing patterns of p–p interactions and the behaviour of floral visitors (e.g. Bluthgen et al. 2006, Petanidou et al. 2008, Kallimanis et al. 2009, Ponisio et al. 2017). It is true that we should have attributed these statements to specific studies (viz. Petanidou et al. 2008, Kallimanis et al. 2009), nevertheless the new version of the text does not include them.

L173-174: how does this indicate keystone species? I think this needs to be explained.

Authors: We used the term “keystone” based on the definition by Power *et al.* (1996) stating that keystone species are those ‘*whose effect is large, and disproportionately large relative to their abundance*’. However, we understand that the discussion on keystone species has been long (Davic 2003), and we would not want to disorientate the readers. Thus, we now replaced the term with “highly influential”. However, specific topological roles that in reality hold the network together (viz. network hubs and connectors) could be finely characterized as “keystone” species for pollination.

L259: missing word? “including IN other”?

Authors: The sentence has been changed. Please see revised version, L284.

L288: run-on sentence

Authors: The sentence has been changed. Please see revised version, L326-328.

L307: What is meant by “evolutionary trends”? This is used repeatedly but I felt never really explained.

Authors: The Reviewer is right. The term was perhaps used loosely. In the revised manuscript, it is used once (L337) referring to the results of the phylogenetic analyses (L133-163).

L308: What is meant by “ecological interdependence” – I didn’t think this was measured here (wouldn’t this require measures of fitness contributions?)

Authors: The partnership of bees and flowering plants is one of the best examples of mutualism both regarding ecology (thus ecological interdependence) and evolution. We felt it was not necessary to explain this in the text. However, please note that this sentence is not part of the new version of the Conclusions.

L314: How have you demonstrated facilitative effects, or even differentiated between facilitative and competitive effects?

Authors: Demonstrating facilitation or competition was not the scope of the present study. These processes are mentioned in the context of implications regarding conservation or restoration approaches.

Methods

L348: I think you mean day-of-year rather than Julian day, which is a continuous measure of time since the Julian Period.

Authors: Thank you for this observation. We meant “day numbers of the Julian calendar”. Changes have been made throughout the text and the Supplementary Material.

Supplemental

L38: 46.7 hours of sampling over two years is not very much– do you have evidence that this was sufficient to capture the rarer species (can you do rarefaction analyses or similar to demonstrate this?).

Authors: Following the Reviewer’s suggestion, we did a rarefaction analysis in order to estimate sampling completeness, using the approach of Chacoff et al. (2012). We find that our effort captured 73.0% of the pollinator species. This proportion is typical for studies of pollination networks (e.g. Chacoff et al. 2012, Trojelsgaard et al. 2013), especially in the region (Petanidou et al. 2008, Nielsen et al. 2011). We added the accumulation curves of pollinator species richness in the Supplementary Information (Fig. S4), and relevant lines of text (revised version, L412-414).

L69: Are there studies you can cite to demonstrate that a single sampling point for floral scents was adequate?

Authors: In comparative studies at phylogenetic (Levin et al. 2003) or community levels (Filella et al. 2013), tradeoffs between sampling breadth and intensity result in “chemical snapshots”, in which relatively few floral scent samples serve as placeholders for populations or species. This leads to the question of whether such sampling omits important variation among individual plants, and whether pollinators can distinguish between such variants. In the absence of discrete chemical polymorphism, honey bees generally cannot distinguish between individual-level variation in scent composition among members of the same populations or horticultural cultivars of snapdragon or mustard flowers when trained under various associative learning paradigms (Wright et al. 2002). However, conditioned honey bees can distinguish between individual plants from different snapdragon cultivars (Wright et al. 2005) or discrete chemotypes of marjoram (Beker et al. 1989) whether tethered or in free flight, when ratios of the same compounds differ significantly between

cultivars using discriminant analysis. Our scent sampling encompassed four replicate scent samples per species, on average, not one; this has been clarified in the updated Methods section (L509-510).

Regarding floral reflectance, Dyer et al. (2007) showed that for snapdragon flowers (*Antirrhinum majus*, including the *mixta* and *nivea* mutants, in controlled laboratory conditions, bumblebees treat wild-type flowers as the same colour, thus any small variations in spectra for wild-type are perceived as the same by the biologically relevant pollinator. The study also shows that single gene mutations (*mixta* or *nivea*) do cause a change in flower colouration that can be perceived by a bee. Thus, the interspecific range of reflectance spectra for this flower is sufficiently low so as not to be perceived as different by a bee (with absolute conditioning). Furthermore, there is evidence that absolute conditioning is a valid psychophysics principle for bee flower constancy behaviour (Dyer 2006); this principle of using a single spectral curve works for describing spectra (species) for large datasets (Chittka and Menzel 1992, Dyer et al. 2012, Shrestha et al. 2013).

Given all the above, we are confident that our sampling effort captured sufficiently the interspecific variation of the scent and of the reflectance of the flowers in the community.

L156,157,158: The citations numbered 30, 31, 32 do not appear in the list of References (the list ends with citation 29).

Authors: This was our mistake. This paragraph, along with the entire Supplementary Methods section, has been moved to Methods. The references have now been added correctly.

L156-157: K is a metric that provides a measure of the strength of signal relative to a model of trait evolution governed by Brownian motion. My understanding is that K should be interpreted with regard to how close the value is to 1 (which corresponds to BM) and that K is sensitive to polytomies in incompletely resolved trees, which seem to be present in your pollinator trees (please see Davies et al. 2012. Incompletely resolved phylogenetic trees inflate estimates of phylogenetic conservatism. *Ecology* 93:242-247 for a way to correct for this).

Authors: It is true that Blomberg's K has been shown to inflate in case of polytomies (Davies et al. 2012), and we thank you for pinpointing this. However, please note that we did not use Blomberg's K , but K^* , which is a different metric with different behavior than K (Pavoine and Ricotta 2013). Yet, to avoid any doubts, we replaced Blomberg's K^* with Pagel's λ (Pagel 1999), which has been shown to be robust against polytomies (Molina-Venegas and Rodríguez 2017). Apart from indices, conclusions about the visitation of the different pollinator groups in phylogenetically related plants can be drawn by simply looking at the mapped variables within the phylogram of the plant community in Fig. S1-S2; the indices (Blomberg's K^* or Pagel's λ) were used in order to statistically support and verify the observations. Using λ is an advantage also because our PGLS models use the phylogenetic correlation matrix based on λ , produced by the function 'corPagel' in the R package *ape* (see revised Methods, L680-685).

References

- Anacker, B. L., J. N. Klironomos, H. Maherali, K. O. Reinhart, and S. Y. Strauss. 2014. Phylogenetic conservatism in plant-soil feedback and its implications for plant abundance. *Ecol Lett* **17**:1613-1621.
- Ayasse, M., F. P. Schiestl, H. F. Paulus, F. Ibarra, and W. Francke. 2003. Pollinator attraction in a sexually deceptive orchid by means of unconventional chemicals. *Proceedings of the Royal Society of London. Series B: Biological Sciences* **270**:517-522.
- Balbuena, J. A., R. Míguez-Lozano, and I. Blasco-Costa. 2013. PACo: A Novel Procrustes Application to Cophylogenetic Analysis. *PLoS ONE* **8**:e61048.
- Barabasi, A. L., and R. Albert. 1999. Emergence of scaling in random networks. *Science* **286**:509-512.
- Beker, R., A. Dafni, D. Eisikowitch, and U. Ravid. 1989. Volatiles of two chemotypes of *Majorana syriaca* L. (Labiatae) as olfactory cues for the honeybee. *Oecologia* **79**:446-451.
- Bluthgen, N., F. Menzel, and N. Bluthgen. 2006. Measuring specialization in species interaction networks. *Bmc Ecology* **6**:9.
- Chacoff, N. P., D. P. Vázquez, S. B. Lomáscolo, E. L. Stevani, J. Dorado, and B. Padrón. 2012. Evaluating sampling completeness in a desert plant-pollinator network. *Journal of Animal Ecology* **81**:190-200.
- Chittka, L., and A. Briscoe. 2001. Why Sensory Ecology Needs to Become More Evolutionary — Insect Color Vision as a Case in Point. Pages 19-37 in F. G. Barth and A. Schmid, editors. *Ecology of Sensing*. Springer Berlin Heidelberg, Berlin, Heidelberg.
- Chittka, L., and R. Menzel. 1992. The Evolutionary Adaptation of Flower Colors and the Insect Pollinators Color-Vision. *Journal of Comparative Physiology a-Sensory Neural and Behavioral Physiology* **171**:171-181.
- Davic, R. D. 2003. Linking keystone species and functional groups: A new operational definition of the keystone species concept - Response. *Conservation Ecology* **7**.
- Davies, T. J., N. J. B. Kraft, N. Salamin, and E. M. Wolkovich. 2012. Incompletely resolved phylogenetic trees inflate estimates of phylogenetic conservatism. *Ecology* **93**:242-247.
- Dyer, A. G. 2006. Discrimination of flower colours in natural settings by the bumblebee species *bombus terrestris* (Hymenoptera : Apidae). *Entomologia Generalis* **28**:257-268.
- Dyer, A. G., S. Boyd-Gerny, S. McLoughlin, M. G. P. Rosa, V. Simonov, and B. B. M. Wong. 2012. Parallel evolution of angiosperm colour signals: common evolutionary pressures linked to hymenopteran vision. *Proceedings of the Royal Society B: Biological Sciences* **279**:3606-3615.
- Dyer, A. G., H. M. Whitney, S. E. J. Arnold, B. J. Glover, and L. Chittka. 2007. Mutations perturbing petal cell shape and anthocyanin synthesis influence bumblebee perception of *Antirrhinum majus* flower colour. *Arthropod-Plant Interactions* **1**:45-55.
- Fenster, C. B., W. S. Armbruster, P. Wilson, M. R. Dudash, and J. D. Thomson. 2004. Pollination syndromes and floral specialization. *Annual Review of Ecology Evolution and Systematics* **35**:375-403.
- Filella, I., C. Primante, J. Llusia, A. M. Martin Gonzalez, R. Seco, G. Farre-Armengol, A. Rodrigo, J. Bosch, and J. Penuelas. 2013. Floral advertisement scent in a changing plant-pollinators market. *Sci Rep* **3**:3434.
- Gomez, J. M., F. Perfectti, and C. P. Klingenberg. 2014. The role of pollinator diversity in the evolution of corolla-shape integration in a pollination-generalist plant clade. *Philos Trans R Soc Lond B Biol Sci* **369**.
- Hadfield, J. D., B. R. Krasnov, R. Poulin, and S. Nakagawa. 2014. A Tale of Two Phylogenies: Comparative Analyses of Ecological Interactions. *American Naturalist* **183**:174-187.
- Hutchinson, M. C., E. F. Cagua, J. A. Balbuena, D. B. Stouffer, and T. Poisot. 2017a. paco: implementing Procrustean Approach to Cophylogeny in R. *Methods in Ecology and Evolution* **8**:932-940.
- Hutchinson, M. C., E. F. Cagua, and D. B. Stouffer. 2017b. Cophylogenetic signal is detectable in pollination interactions across ecological scales. *Ecology* **98**:2640-2652.
- Johnson, S. D., and R. A. Raguso. 2016. The long-tongued hawkmoth pollinator niche for native and invasive plants in Africa. *Annals of Botany* **117**:25-36.
- Junker, R. R., N. Blüthgen, T. Brehm, J. Binkenstein, J. Paulus, H. Martin Schaefer, and M. Stang. 2013. Specialization on traits as basis for the niche-breadth of flower visitors and as structuring mechanism of ecological networks. *Functional Ecology* **27**:329-341.

- Junker, R. R., N. Hocherl, and N. Bluthgen. 2010. Responses to olfactory signals reflect network structure of flower-visitor interactions. *Journal of Animal Ecology* **79**:818-823.
- Jürgens, A., S. L. Wee, A. Shuttleworth, and S. D. Johnson. 2013. Chemical mimicry of insect oviposition sites: a global analysis of convergence in angiosperms. *Ecol Lett* **16**:1157-1167.
- Kallimanis, A. S., T. Petanidou, J. Tzanopoulos, J. D. Pantis, and S. P. Sgardelis. 2009. Do plant-pollinator interaction networks result from stochastic processes? *Ecological Modelling* **220**:684-693.
- Kamilar, J. M., and N. Cooper. 2013. Phylogenetic signal in primate behaviour, ecology and life history. *Philos Trans R Soc Lond B Biol Sci* **368**:20120341.
- Kantsa, A., R. A. Raguso, A. G. Dyer, S. P. Sgardelis, J. M. Olesen, and T. Petanidou. 2017. Community-wide integration of floral colour and scent in a Mediterranean scrubland. *Nature Ecology & Evolution* **1**:1502-1510.
- Klempay, B., I. Lim, M. Mallula, A. Olsen, K. Park, D. Silva, V. Gonzalez, J. Hranitz, T. Petanidou, and J. Barthell. 2016. The Effect of Introducing Differing Color Floral Morphs on Bee Visitation in a Native Population of *Vitex agnus-castus* on the Greek Island of Lesvos. Pages E316-E316 *in* Integrative and Comparative Biology. OXFORD UNIV PRESS INC JOURNALS DEPT, 2001 EVANS RD, CARY, NC 27513 USA.
- Larue, A. A. C., R. A. Raguso, and R. R. Junker. 2016. Experimental manipulation of floral scent bouquets restructures flower-visitor interactions in the field. *Journal of Animal Ecology* **85**:396-408.
- Lázaro, A., T. Tscheulin, J. Devalez, G. Nakas, and T. Petanidou. 2016a. Effects of grazing intensity on pollinator abundance and diversity, and on pollination services. *Ecological Entomology* **41**:400-412.
- Lázaro, A., T. Tscheulin, J. Devalez, G. Nakas, A. Stefanaki, E. Hanlidou, and T. Petanidou. 2016b. Moderation is best: effects of grazing intensity on plant-flower visitor networks in Mediterranean communities. *Ecological Applications* **26**:796-807.
- Legendre, P., Y. Desdevises, and E. Bazin. 2002. A statistical test for host-parasite coevolution. *Systematic Biology* **51**:217-234.
- Levin, R. A., L. A. McDade, and R. A. Raguso. 2003. The systematic utility of floral and vegetative fragrance in two genera of Nyctaginaceae. *Systematic Biology* **52**:334-351.
- Lunau, K. 2004. Adaptive radiation and coevolution — pollination biology case studies. *Organisms Diversity & Evolution* **4**:207-224.
- Michener, C. D. 2007. The bees of the world. The Johns Hopkins University Press, Baltimore.
- Molina-Venegas, R., and M. Á. Rodríguez. 2017. Revisiting phylogenetic signal; strong or negligible impacts of polytomies and branch length information? *Bmc Evolutionary Biology* **17**:53.
- Nielsen, A., I. Steffan-Dewenter, C. Westphal, O. Messinger, S. G. Potts, S. P. M. Roberts, J. Settele, H. Szentgyörgyi, B. E. Vaissière, M. Vaitis, M. Woyciechowski, I. Bazos, J. C. Biesmeijer, R. Bommarco, W. E. Kunin, T. Tscheulin, E. Lamborn, and T. Petanidou. 2011. Assessing bee species richness in two Mediterranean communities: importance of habitat type and sampling techniques. *Ecological Research* **26**:969-983.
- Olesen, J. M., J. Bascompte, Y. L. Dupont, and P. Jordano. 2007. The modularity of pollination networks. *Proceedings of the National Academy of Sciences of the United States of America* **104**:19891-19896.
- Olesen, J. M., J. Bascompte, H. Elberling, and P. Jordano. 2008. Temporal dynamics in a pollination network. *Ecology* **89**:1573-1582.
- Ollerton, J., R. Alarcon, N. M. Waser, M. V. Price, S. Watts, L. Cranmer, A. Hingston, C. I. Peter, and J. Rotenberry. 2009. A global test of the pollination syndrome hypothesis. *Annals of Botany* **103**:1471-1480.
- Pagel, M. 1999. Inferring the historical patterns of biological evolution. *Nature* **401**:877-884.
- Paradis, E. 2012. Analysis of Phylogenetics and Evolution with R. second edition. Springer, New York.
- Pavoine, S., and C. Ricotta. 2013. Testing for Phylogenetic Signal in Biological Traits: The Ubiquity of Cross-Product Statistics. *Evolution* **67**:828-840.
- Pearse, W. D., M. W. Cadotte, J. Cavender-Bares, A. R. Ives, C. M. Tucker, S. C. Walker, and M. R. Helmus. 2015. pez: phylogenetics for the environmental sciences. *Bioinformatics* **31**:2888-2890.
- Petanidou, T., and W. N. Ellis. 1993. Pollinating Fauna of a Phrygic Ecosystem: Composition and Diversity. *Biodiversity Letters* **1**:9-22.

- Petanidou, T., A. S. Kallimanis, J. Tzanopoulos, S. P. Sgardelis, and J. D. Pantis. 2008. Long-term observation of a pollination network: fluctuation in species and interactions, relative invariance of network structure and implications for estimates of specialization. *Ecology Letters* **11**:564-575.
- Ponisio, L. C., M. P. Gaiarsa, and C. Kremen. 2017. Opportunistic attachment assembles plant–pollinator networks. *Ecology Letters* **20**:1261-1272.
- Power, M. E., D. Tilman, J. A. Estes, B. A. Menge, W. J. Bond, L. S. Mills, G. Daily, J. C. Castilla, J. Lubchenco, and R. T. Paine. 1996. Challenges in the quest for keystones. *Bioscience* **46**:609-620.
- Rafferty, N. E., and A. R. Ives. 2013. Phylogenetic trait-based analyses of ecological networks. *Ecology* **94**:2321-2333.
- Rasmussen, C., Y. L. Dupont, J. B. Mosbacher, K. Trøjelsgaard, and J. M. Olesen. 2013. Strong Impact of Temporal Resolution on the Structure of an Ecological Network. *PLoS ONE* **8**:e81694.
- Renoult, J. P., N. Blüthgen, J. Binkenstein, C. N. Weiner, M. Werner, and H. M. Schaefer. 2015. The relative importance of color signaling for plant generalization in pollination networks. *Oikos* **124**:347-354.
- Saleh, N., A. G. Scott, G. P. Bryning, and L. Chittka. 2007. Distinguishing signals and cues: bumblebees use general footprints to generate adaptive behaviour at flowers and nest. *Arthropod-Plant Interactions* **1**:119-127.
- Schiestl, F. P. 2010. The evolution of floral scent and insect chemical communication. *Ecology Letters* **13**:643-656.
- Schiestl, F. P., and S. Dotterl. 2012. THE EVOLUTION OF FLORAL SCENT AND OLFACTORY PREFERENCES IN POLLINATORS: COEVOLUTION OR PRE-EXISTING BIAS? *Evolution* **66**:2042-2055.
- Searcy, W. A., and S. Nowicki. 2005. *The evolution of animal communication: reliability and deception in signaling systems*. Princeton University Press.
- Shrestha, M., A. G. Dyer, and M. Burd. 2013. Evaluating the spectral discrimination capabilities of different pollinators and their effect on the evolution of flower colors. *Communicative and Integrative Biology* **6**:e24000.
- Swenson, N. G. 2014. *Functional and Phylogenetic Ecology* in R. Springer, New York.
- Trojelsgaard, K., M. Baez, X. Espadaler, M. Nogales, P. Oromi, F. La Roche, and J. M. Olesen. 2013. Island biogeography of mutualistic interaction networks. *Journal of Biogeography* **40**:2020-2031.
- Waser, N. M., and L. A. Real. 1979. Effective mutualism between sequentially flowering plant species. *Nature* **281**:670.
- Wright, G. A., A. Lutmerding, N. Dudareva, and B. H. Smith. 2005. Intensity and the ratios of compounds in the scent of snapdragon flowers affect scent discrimination by honeybees (*Apis mellifera*). *Journal of Comparative Physiology a-Neuroethology Sensory Neural and Behavioral Physiology* **191**:105-114.
- Wright, G. A., B. D. Skinner, and B. H. Smith. 2002. Ability of honeybee, *Apis mellifera*, to detect and discriminate odors of varieties of canola (*Brassica rapa* and *Brassica napus*) and snapdragon flowers (*Antirrhinum majus*). *Journal of Chemical Ecology* **28**:721-740.

REVIEWERS' COMMENTS:

Reviewer #1 (Remarks to the Author):

Thank you for your thorough revision of this interesting paper. I am very happy with the changes you have made and I now feel that your key messages are much clearer. I feel this paper now makes an excellent contribution to the literature and I have no additional comments.

Reviewer #2 (Remarks to the Author):

The authors went through a thorough revision of the manuscript. The rearrangement of the introduction is helpful and I agree with the authors that the major scope (and novelty) of the approach has been (and should be) analysing (or developing tools for analysing) the role floral phenotype on pollination networks. The authors have now added additional results to highlight this point. The introduction and discussion is now well balanced in terms of the goals and topics that are covered. The authors have also improved language and style of the whole manuscript a bit – although I found the paper was already very well written in its original version. The authors have addressed some statistical questions/concerns: To verify their statistical analysis Pagel's λ was added. Furthermore, they also re-ran the multivariate-response generalized linear and the PGLS models without *Heliotropium lasiocarpum* because due to the late blooming of the species the method might underestimate the species apparency.

I like the extra/modified paragraph in the results and discussion section (L336-355) on the interplay of the ecological and evolutionary trends. There are some very interesting outcomes/hypotheses that could be tested in the future. For instance, that "...generalized behaviours of pollinators might include a somewhat cryptic specialization for floral sensory stimuli." This challenges the existing framework on flower visitor specialization because it highlights the role of the sensory system of animals as a possible driver for specialization. I have noticed that the authors have made an effort to address the concerns of one reviewer regarding their conclusions that touch on the role of sensory systems. However, I found the discussion on the putative role of floral stimuli for the formation of ecological interaction-networks and the underlying evolutionary trends the most interesting part of the paper (maybe because my background is chemical ecology and sensory ecology). I am therefore very glad that the authors added some more background information which explains now a bit better how they derive their conclusions. At the same time they are particularly careful to emphasise that their research is only a first step. Yes, it is a first step but considering the enormous amount of work that is needed to establish a framework for using sensory data in the context of a network approach this is a very important one.

Sincerely yours
Andreas Jürgens

Reviewer #3 (Remarks to the Author):

I read the authors' responses to my own comments (and out of interest, those of the other reviewers) carefully, along with the revised ms. I think the authors have adequately addressed my comments and am glad the revised phylogenetic analyses yielded additional insights.